# IPO: Interpretable Prompt Optimization for Vision-Language Models

**Yingjun Du[1*], Wenfang Sun[2*], Cees G. M. Snoek[1]**

[1]AIM Lab, University of Amsterdam  [2]University of Science and Technology of China

## Abstract

Pre-trained vision-language models like CLIP have remarkably adapted to various downstream tasks. Nonetheless, their performance heavily depends on the specificity of the input text prompts, which requires skillful prompt template engineering. Instead, current approaches to prompt optimization learn the prompts through gradient descent, where the prompts are treated as adjustable parameters. However, these methods tend to lead to overfitting of the base classes seen during training and produce prompts that are no longer understandable by humans. This paper introduces a simple but interpretable prompt optimizer (IPO), that utilizes large language models (LLMs) to generate textual prompts dynamically. We introduce a Prompt Optimization Prompt that not only guides LLMs in creating effective prompts but also stores past prompts with their performance metrics, providing rich in-context information. Additionally, we incorporate a large multimodal model (LMM) to condition on visual content by generating image descriptions, which enhance the interaction between textual and visual modalities. This allows for the creation of dataset-specific prompts that improve generalization performance, while maintaining human comprehension. Extensive testing across 11 datasets reveals that IPO not only improves the accuracy of existing gradient-descent-based prompt learning methods but also considerably enhances the interpretability of the generated prompts. By leveraging the strengths of LLMs, our approach ensures that the prompts remain human-understandable, thereby facilitating better transparency and oversight for vision-language models.

## 1   Introduction

Vision-language models, trained on a diverse array of image-text pairs encapsulating a broad vocabulary of real-world concepts [1, 2, 3], have demonstrated notable adaptability across various downstream tasks [4, 5, 6, 7]. These models perform zero-shot image classification by filling in a predefined prompt template (e.g., "`a photo of a [CLASS]`") with specific class names for the text encoder. Despite their effective generalization to new tasks, the performance can be influenced by minor changes in the wording of prompt templates [8]. Instead of manually creating hand-crafted prompts, recent developments in natural language processing [9, 10] and computer vision [8, 11, 12, 13] have proposed methods to learn a set of soft prompts with minimal labeled data. Despite the strides made in learning prompts, the current state of the art remains limited by its lack of interpretability and the overfitting problems on the base classes, which can be prohibitive in diverse and dynamic application environments. These limitations underscore the need for a more adaptable and user-friendly approach to prompt optimization in vision-language models.

Drawing from recent advancements in using large language models (LLMs) as optimization tools [14], our paper, for the first time, incorporates these capabilities into vision-language modeling. Unlike gradient descent-based methods [8, 11, 13], which often fail to provide explanations for the generated

---

[*]Equal contribution.

38th Conference on Neural Information Processing Systems (NeurIPS 2024).

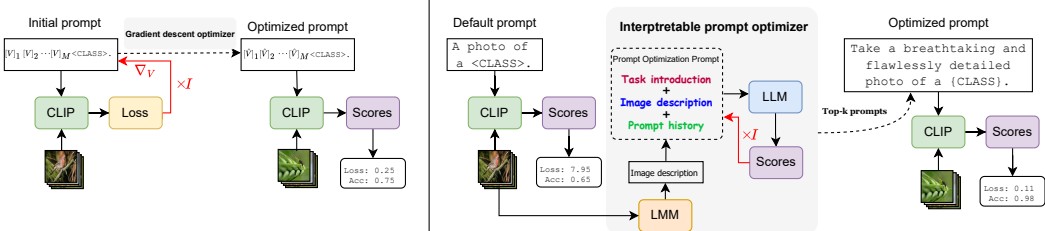

(a) Gradient-based prompt optimization.    (b) Interpretable prompt optimization.

**Figure 1:** Comparison between traditional gradient-based prompt optimization (a) and our interpretable prompt optimization (b) for vision-language models. Traditional gradient descent-based prompt learning methods [8, 11] treat the text prompt as learnable parameters $V$. By minimizing the loss through gradient descent on the training set, an optimized prompt $\hat{V}$ is obtained after $I$ iterations, which is not interpretable by humans. In contrast, our interpretable prompt optimization leverages an LLM as optimizer to optimize the loss and accuracy. After $I$ iterations, the resulting optimized top prompt is effective and human-readable.

prompts and tend to overfit on base classes, natural language-based methods enable LLMs to develop and refine solutions through continuous feedback iteratively. This approach improves interpretability in complex tasks like prompt optimization for vision-language models, making it easier for humans to understand the generated prompts. However, existing research on these methods [14, 15, 16] primarily addresses language tasks and has not yet explored their potential for integrating prompt optimization with an LLM in vision-language models.

To address these challenges, this paper proposes an interpretable prompt optimizer (IPO) for vision-language models that leverages the capabilities of LLMs to generate and refine text prompts dynamically. First, we design a Prompt Optimization Prompt to prompt LLMs to generate more effective prompts that improve the accuracy of CLIP and reduce the loss in base classes. Our Prompt Optimization Prompt also stores past prompts along with their corresponding accuracy and loss as episodic memory, thereby providing richer in-context information to enable LLMs to generate more effective prompts. Second, to incorporate image information within the Prompt Optimization Prompt, we propose using a large multimodal model (LMM) to generate descriptions of images in base classes that can be added to the Prompt Optimization Prompt. This integration facilitates a more intuitive interaction between the textual and visual modalities, which allows the Prompt Optimization Prompt to utilize image information, thereby generating dataset-specific prompts to enhance the generalization performance of CLIP. The framework of our IPO, illustrated in Figure 1, showcases the comparison between traditional gradient-based prompt optimization and the proposed interpretable prompt optimization. Third, the prompts generated by our optimizer are human-interpretable. For example, on the Food101 dataset [17], the initial prompt evolves from [“a photo of a [CLASS]”] to [“Categorize the image depicting a delicious and appetizing <CLASS> with remarkable visual qualities.”]. Our generated prompts perform 10.29% better in novel classes than the gradient-based method CoOP [8], reducing overfitting while maintaining interpretability.

We validated our IPO across 11 different datasets, demonstrating that it surpasses traditional gradient-based state-of-the-art methods in accuracy and excels in interpretability. Our approach generates human-comprehensible prompts that can be seamlessly integrated into existing vision-language models to enhance performance. We conducted rigorous comparative experiments to quantify the interpretability between gradient-based prompt learning and our method. We demonstrated the importance of specific keywords in our generated prompts and revealed that not all tokens learned through traditional prompt learning methods are essential.

## 2 Related work

**Prompt learning in vision-language models.** Prompt learning, originally introduced in the natural language processing community [18, 19, 20], involves applying a fixed function to input tokens to provide task instructions to the model. In the computer vision community, prompt learning has been explored in various forms, including textual prompt tuning [8, 11, 21, 22, 23, 24], and prefix tuning [13, 12, 25, 26, 27, 28, 29]. 1) Prompt tuning mainly involves treating text prompts as learnable parameters, using a small amount of data to fine-tune these parameters. As pioneered by CoOp [8]

and CoCoOp [11], which both fine-tune a CLIP vision-language model [30] for few-shot transfer by optimizing a continuous set of prompt vectors within its language branch. Bayesian prompt learning [21] formulated prompt learning as a variational inference problem and demonstrated its ability for unseen class generalization. 2) Prefix tuning primarily involves adding learnable tokens to the text encoder [31], vision encoder [12, 25], or both encoders [13, 27, 28, 32]. These tokens are fine-tuned using a small amount of data. Note that these methods do not optimize the initial text prompts. Instead, they focus on enhancing the model's understanding capabilities by integrating these additional, trainable tokens. Our method belongs to prompt tuning, but unlike previous approaches that use gradient descent to optimize prompts, we propose using LLMs to optimize prompts. Our method leverages the natural language capabilities of LLMs to iteratively refine feedback-based prompts, aiming to enhance both the effectiveness and the explainability of the prompts.

**LLMs as prompt optimizers.** Several recent works explore the role of LLMs as prompt optimizers for NLP tasks. Some use LLMs to directly optimize the task instruction for in-context learning [33, 34, 14]. Other studies use LLMs to mutate prompts for evolutionary algorithms [35, 36]. However, to the best of our knowledge, no existing studies have investigated how LLMs could be used to optimize text prompts within vision-language models. This approach could potentially open up new avenues for integrating and enhancing the capabilities of vision-language models through more effective and contextually appropriate text prompts.

**Meta-prompting.** Suzgun and Kalai [37] introduce meta-prompting to transform a single LLM into a versatile "conductor" capable of managing and integrating multiple independent LLM queries. By using high-level instructions, meta-prompting guides the LLM in decomposing complex tasks into smaller subtasks. The core of OPRO [14] involves designing a meta-prompt for LLMs to optimize prompts for each task. This meta-prompt includes two key pieces of information: previously generated prompts with their corresponding training accuracies and a description of the optimization problem. Self-select [38] leverages meta-prompting to optimize instruction selection. It considers a set of provided templates and chooses the most suitable template. Meta-prompting is related to instruction tuning [39] as both techniques provide high-level guidance to improve the performance and adaptability of LLMs. However, while instruction tuning focuses on fine-tuning models with a variety of tasks to improve generalization, meta-prompting offers the advantage of dynamically guiding the model to decompose and manage complex tasks in real-time. Liu et al. [40] proposes a method that utilizes LLMs as black-box optimizers for vision-language models, iteratively refining prompts based on in-context examples. Their approach focuses on leveraging ChatGPT to improve prompt templates for visual classification tasks. Mirza et al. [41] explores a different aspect of prompt optimization by focusing on zero-shot vision-language models. Our Prompt Optimization Prompt is akin to meta-prompting, it stores past prompts along with their corresponding accuracy and loss, thereby providing richer in-context information to enable LLMs to generate more effective prompts. Different from prior meta-prompting, our Prompt Optimization Prompt generates prompts beyond LLMs for vision-language models.

## 3 Preliminaries

**Contrastive Language-Image Pre-Training (CLIP).** The goal of CLIP [30] is to develop an image encoder $f_I$ and a text encoder $g_T$ via contrastive pre-training with a large collection of paired images and captions. This process aims to map image-text pairs to a common semantic space. After the pre-training phase, CLIP is able to perform zero-shot visual recognition by treating classification as a task of matching images to text. Specifically, the placeholder term "`[CLASS]`" is used within a prompt template (e.g., "`a photo of a [CLASS]`") for the text encoder $g_T$. Here, $g_T(\mathbf{T}_i)$ denotes the text features adapted for class $i$, and the probability of classifying class $i$ from an image $\mathbf{I}$ is:

$$p(y=i|\mathbf{I}) = \frac{\exp(\langle g_T(\mathbf{T}_i), f_I(\mathbf{I})\rangle/\tau)}{\sum_{j=1}^{K} \exp(\langle g_T(\mathbf{T}_j), f_I(\mathbf{I})\rangle/\tau)}, \tag{1}$$

where $\langle g_T(\mathbf{T}_i), f_I(\mathbf{I})\rangle$ represents the cosine similarity between the image feature $f_I(\mathbf{I})$ and the text feature $g_T(\mathbf{T}_i)$ specific to the $i$-th class, $K$ is the total number of classes, and $\tau$ is the temperature parameter that is tuned during training.

**Prompt learning** improves the adaptability of the CLIP model by eliminating the need for manual prompt engineering. It facilitates the automatic generation of prompts using a limited number of examples from a downstream task. CoOp [8] presents a method where a set of $M$ continu-

ous context vectors $\boldsymbol{V} = \{\boldsymbol{v}_1, \boldsymbol{v}_2, \ldots, \boldsymbol{v}_M\}$ serve as the learnable prompt. The constructed prompt $\boldsymbol{T}_i = \{\boldsymbol{v}_1, \boldsymbol{v}_2, \ldots, \boldsymbol{v}_M, \boldsymbol{c}_i\}$ merges these learnable context vectors $\boldsymbol{V}$ with the class-specific token embedding $\boldsymbol{c}_i$, which is then processed by the text encoder $g_T(\cdot)$. In CoOp, the optimization of these static context vectors $\boldsymbol{V}$ aims to minimize the negative log-likelihood for the correct class token:

$$\mathcal{L}_{\text{CE}}(\boldsymbol{V}) = - \sum_i \boldsymbol{y}_i \log p(\boldsymbol{T}_i | \boldsymbol{I}), \tag{2}$$

here, $\boldsymbol{y}_i$ represents the one-hot encoded ground-truth label for class $i$. In downstream applications, the pre-trained model parameters are kept unchanged, which allows the learnable prompt vectors $\boldsymbol{V}$ to be optimized efficiently using only a small number of samples through the minimization of the cross-entropy loss.

## 4 Methods

Figure 1 depicts the comprehensive structure of our interpretable prompt optimizer. At each step of the optimization, the LLM generates candidate prompts for the vision-language task by considering both the description of the optimization problem and the feedback from previously evaluated prompts stored in the Prompt Optimization Prompt. These new prompts are then assessed and incorporated into the Prompt Optimization Prompt for future optimization cycles. The optimization process concludes either when the LLM can no longer generate prompts that improve the optimization scores, or when a predefined maximum number of optimization steps is reached. Next, we will detail the design of the Prompt Optimization Prompt and explain how image information is integrated into the Prompt Optimization Prompt.

**Prompt Optimization Prompt design** At the core of our optimizer is the design of the Prompt Optimization Prompt, which enhances the performance of the vision-language model by optimizing the prompts through the *prompt* LLM. Figure 2 shows an example of our Prompt Optimization Prompt. Our Prompt Optimization Prompt consists of the following components: (1) Instructions: These guide the LLM by clearly defining its task to optimize the prompt for achieving better performance in classification tasks. (2) Textual descriptions of training images: These descriptions provide the LLM with detailed information about the images, enabling it to generate dataset-specific prompts. (3) Previously generated prompts and corresponding scores: This component supplies in-context information, including past prompts and their performance metrics, allowing the LLM to refine its prompt generation more accurately. By incorporating these elements, our approach leverages the iterative refinement capabilities of LLMs to dynamically generate and optimize text prompts. The instructions ensure that the LLM understands the optimization goal, the textual descriptions offer rich image-related context, and the historical data aids in producing more effective and precise prompts.

**Textual descriptions of training images** For the textual descriptions of training images, we utilize a large multimodal model (LMM) to generate text descriptions for each training image. Specifically, we employ MiniCPM-V-2.0 [43] to generate descriptions of the content of images from base classes. In the appendix, we provide content descriptions for some images from each dataset generated using MiniCPM-V-2.0. We denote the extracted image textual features as $f_M(\cdot)$.

Additionally, we have attempted to directly optimize prompts using the LMM with the Prompt Optimization Prompt. Specifically, we input images from the base classes and the Prompt Optimization Prompt into the LMM, aiming for the LMM to generate better prompts. We experimented with six different LMMs: BLIP-2 [44], Qwen-VL-Chat-9.6B [45], FUYU-8B [46], MiniCPM-V-2.0 [43], and llava-llama3-8B [47]. Unfortunately, all six models failed to understand our Prompt Optimization Prompt and generated new prompts that were merely descriptions of the images, not the universal prompts we desired. This failure might be due to the fact that the training of these LMMs did not consider such a task. Note that image descriptions are not mandatory. In our 16-shot experiments, we omitted the image descriptions in the Prompt Optimization Prompt due to the limited text input length that the LLM can handle.

**Episodic memory retrieval** We utilize an episodic memory mechanism to retrieve past prompts and their corresponding scores, which include metrics such as loss and accuracy. Here, we denote the memory as $\mathcal{M}$. During each iteration, we retrieve the top-20 prompts $\mathcal{R}(\mathcal{M})$, based on their accuracy from $\mathcal{M}$ and use them as the current memory, denoted as $\mathbf{m}$. Moreover, we consistently include the prompt "a photo of <CLASS>" in our history at every step, as this is a frequently used

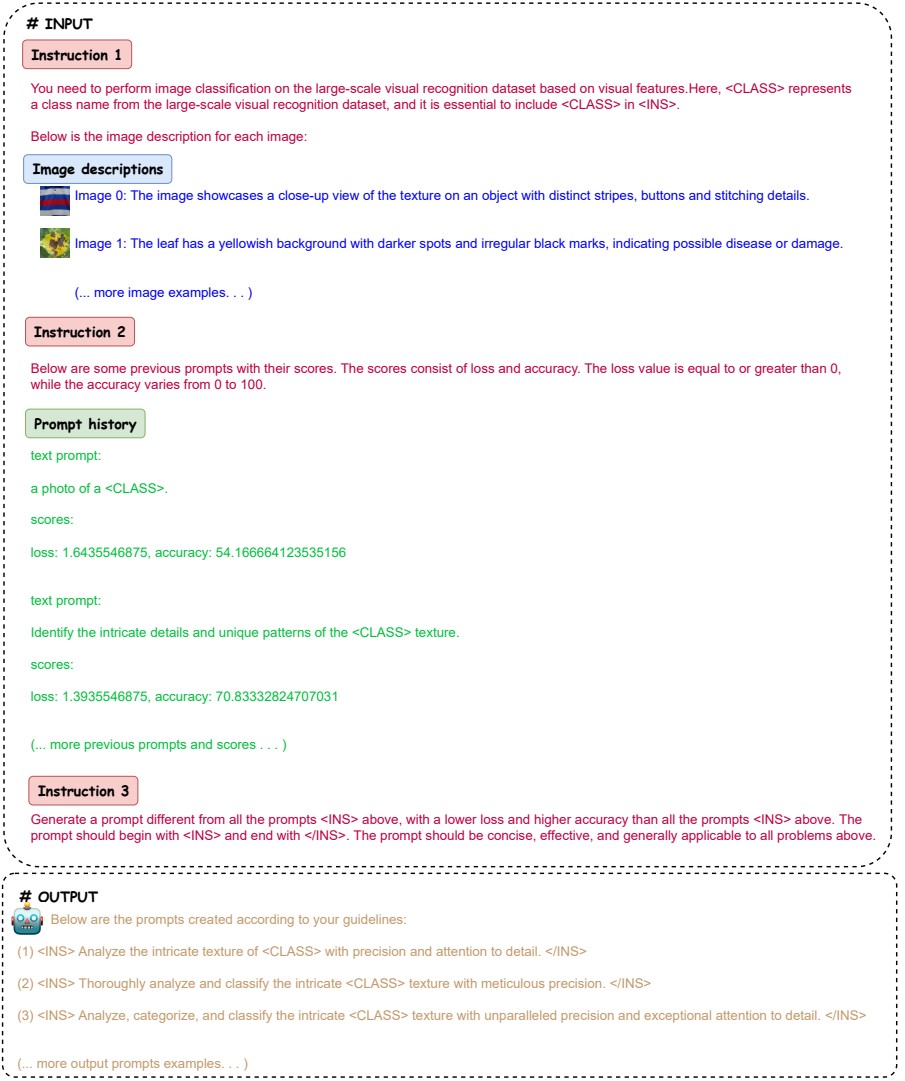

**Figure 2:** An example of our Prompt Optimization Prompt with input and output on the DTD [42] dataset. The **red** text represents instructions given to the large language model, the **blue** text denotes the image descriptions generated by the large multimodal model, and the **green** text indicates the top-20 previously generated prompts retrieved from episodic memory along with their corresponding scores. **yellow** indicates the output prompt.

and effective prompt within the CLIP framework [30]. Therefore, our optimization loss is defined as:

$$\mathcal{L}_{\text{CE}} = -\sum_i \boldsymbol{y}_i \log p(\hat{\boldsymbol{T}}_i | f_M(I), \mathbf{m}, \mathcal{I}), \tag{3}$$

where $\mathcal{I}$ indicates the our designed instruction for LLM, $\hat{\boldsymbol{T}}$ represents the new human-interpretable text prompt optimized by the LLM. Note that our optimizer is parameter-free, which differentiates it from traditional gradient-based prompt learning methods. Instead, we leverage the LLM to optimize the prompt, reducing $\mathcal{L}_{\text{CE}}$ iteratively until convergence.

The input and output example in Figure 2 shows the structured information fed into the LLM, while output demonstrates the optimized prompts generated by the LLM. For more detailed examples of Prompt Optimization Prompt input and output, please refer to the appendix.

# 5 Experiments

## 5.1 Experimental setup

We validate the effectiveness of our approach on the base-to-new generalization benchmark for evaluating prompt learning in vision-language models [8, 11]. Across all experiments, we benchmark the models' performance in a 1-shot and commonly used 16-shot setting. To ensure consistency, all results from learning-based methods are averaged over three random seeds. We use the harmonic mean (H) as the average metric, which is a common approach in prompt learning for vision-language models.

**Eleven Datasets.** We follow CLIP [30] and CoOp [8] to use 11 image classification datasets, *i.e.*, ImageNet [48] and Caltech101 [49] for generic object classification, OxfordPets [50], Stanford-Cars [51], Flowers102 [52], Food101 [17] and FGVCAircraft [53] for fine-grained image recognition, EuroSAT [54] for satellite image classification, UCF101 [55] for action classification, DTD [42] for texture classification, and SUN397 [56] for scene recognition.

**Six Baselines.** To conduct a comparative evaluation, we utilize a number of established baselines including CLIP [30], Coop [8], CoCoOp [11], MaPLe [13], PromptSRC [28], and CoPrompt [32]. Note that all methods do not present 1-shot results in their publications, so we perform 1-shot experiments using their available code.

**Training details.** We use GPT-3.5 Turbo as our default optimizer, iterating 100 steps for each dataset to derive the final prompt. At each step, we generate five prompts and compare their accuracy with past prompts, storing the top-20 prompts in our history. Ultimately, we select the prompt with the highest accuracy as the final prompt. For generating image descriptions, we employ MiniCPM-V-2.0 [43] as the default LMM, using the prompt: "[Please generate the description in detail, do not provide the class name in the description.]". We added image descriptions to the 1-shot Prompt Optimization Prompt but not to the 16-shot version due to the character input limitations of GPT-3.5 Turbo, which prevent adding detailed information for each class's images. All experiments were conducted on a GeForce RTX 3090. Our code is available at https://github.com/lmsdss/IPO.

## 5.2 Results

**Interpretable prompt analysis.** To analyze the importance of specific words or phrases in text prompts, we display a comparative experiment on the Flower102 dataset. Specifically, in Table 1a, we use "a photo of a <CLASS>" as the prompt, which is the most commonly utilized format. By employing occlusion sensitivity analysis, we individually remove each word to test the importance of the remaining words. We find that using just "<CLASS>" as the prompt performs only 0.89% lower on the novel classes than "a photo of a <CLASS>", indicating that the CLIP model can generate more discriminative features using just the category name. Additionally, the prompt "a photo <CLASS>." achieves the best performance. By comparing "<CLASS>" with "photo <CLASS>", we determine that the word *photo* is particularly significant on the novel classes.

In Table 1b, we present the results of the CoOP and CoCoOP models when some learned prompt tokens are removed to analyze which token is most crucial. Surprisingly, for both models, performance improves in novel classes when some or all tokens are removed. This indicates severe overfitting in base classes by these models, where the learned tokens are only applicable to base classes. Removing all tokens allows the models to retain some of the original performance of CLIP on novel classes. Additionally, these methods, which only learn tokens, make it challenging for humans to understand the specific meanings of each token, complicating interpretation.

In Table 1c, we also demonstrate the final prompt produced by our model: "Identify the unique visual features of the <CLASS> flower accurately," which achieves a performance of 79.6%, surpassing the original CLIP by 2.9% on novel classes. When comparing this optimal prompt with others, removing any words from it results in worse performance than the original prompt. Specifically, comparing "Identify <CLASS>" with "<CLASS>" reveals that including "Identify" boosts performance by 1.7% on novel classes. This highlights the importance of the word "Identify" in datasets like Flower102. We have included additional analyses on various datasets in the appendix.

Overall, the comparative experiments demonstrate that our prompt can be more easily interpreted and understood by humans, while also providing insights into the significance of certain key words

| Prompts | CLIP [30] | | |
|---|---|---|---|
| | Base | Novel | H |
| a photo of a <CLASS>. | **69.34** | 76.72 | **72.84** |
| <CLASS>. | 61.75 | 75.83 | 68.07 |
| a <CLASS>. | 63.04 | 75.02 | 68.51 |
| photo <CLASS>. | 64.72 | 76.74 | 70.22 |
| of <CLASS>. | 63.02 | 76.17 | 68.97 |
| a photo <CLASS>. | 66.52 | **77.25** | 71.48 |
| photo of <CLASS>. | 61.33 | 77.12 | 68.32 |
| of a <CLASS>. | 63.37 | 75.91 | 69.08 |
| a photo of <CLASS>. | 59.15 | **77.25** | 70.00 |
| photo of a <CLASS>. | 69.16 | 76.72 | 72.74 |

**(a)** CLIP

| Prompts | CoOP [8] | | | CoCoOP [11] | | |
|---|---|---|---|---|---|---|
| | Base | Novel | H | Base | Novel | H |
| Token 1, 2, 3, 4 | 71.47 | 72.47 | 71.97 | 73.67 | 75.50 | 74.57 |
| None | 61.75 | **75.83** | 68.07 | 61.75 | 75.83 | 68.07 |
| Token 1 | 76.02 | 73.38 | **74.68** | 75.83 | 76.92 | 76.37 |
| Token 2 | 70.07 | 74.26 | 72.10 | **76.75** | 76.98 | 76.86 |
| Token 3 | 69.62 | 72.88 | 71.21 | 76.74 | 77.27 | 77.00 |
| Token 4 | 70.85 | 67.75 | 69.27 | 76.51 | 77.64 | 77.07 |
| Token 1, 2 | 75.04 | 74.23 | 74.63 | 76.02 | 77.23 | 76.62 |
| Token 3, 4 | 70.13 | 66.75 | 68.40 | 76.07 | **78.75** | **77.39** |
| Token 1, 4 | 75.92 | 68.81 | 72.20 | 76.49 | 78.22 | 77.35 |
| Token 1, 2, 3 | 72.55 | 74.04 | 73.29 | 76.51 | 75.53 | 76.02 |
| Token 1, 2, 4 | 74.82 | 70.52 | 72.61 | 76.04 | 77.92 | 76.97 |
| Token 1, 3, 4 | **76.25** | 66.15 | 70.84 | 75.53 | 78.37 | 76.92 |
| Token 2, 3, 4 | 70.91 | 67.93 | 69.39 | 75.25 | 77.71 | 76.46 |

**(b)** CoOP and CoCoOP.

| Prompts | IPO | | |
|---|---|---|---|
| | Base | Novel | H |
| Identify the unique visual features of the <CLASS> flower accurately. | **74.17** | **79.65** | **76.81** |
| <CLASS>. | 61.75 | 75.83 | 68.07 |
| Visual features of the <CLASS>. | 64.42 | 74.64 | 69.15 |
| Identify unique visual features <CLASS>. | 63.28 | 77.21 | 69.55 |
| Identify the unique visual features of the <CLASS> accurately. | 65.67 | 77.35 | 71.03 |
| The unique visual features <CLASS> flower. | 66.42 | 76.67 | 71.18 |
| Identify the unique <CLASS>. | 67.22 | 77.73 | 72.09 |
| Identify the <CLASS>. | 68.88 | 76.35 | 72.42 |
| Identify <CLASS>. | 69.53 | 77.52 | 73.31 |
| Features of the <CLASS> flower accurately. | 72.03 | 77.82 | 74.81 |
| Visual features of the <CLASS> flower. | 71.93 | 78.55 | 75.09 |
| The unique visual features of the <CLASS> flower accurately. | 72.67 | 78.72 | 75.57 |
| Identify the unique visual features of the <CLASS> flower. | 73.82 | 79.11 | 76.37 |

**(c)** *This paper*: IPO

**Table 1:** Comparison of various prompts with occlusion sensitivity analysis across different models on the Flower102 dataset [52]. The shaded areas in the table indicate the original performance of each method. Bold blue refers to the result of the best prompt for each model. The original CLIP model shows particular sensitivity to the word *photo*. In contrast, the tokens learned by CoOP and CoCoOP affect especially base class performance, while removing these learned tokens improves novel class performance. By contrast, with our interpretable prompt optimization, every word makes a meaningful contribution to both base and new classes. We provide results for more datasets in the appendix.

| Models | Base | Novel | H |
|---|---|---|---|
| Phi2-2.7B | 71.15 | 75.43 | 73.22 |
| PaLM 2-L | 71.32 | 75.93 | 73.55 |
| PaLM 2-L-IT | 71.13 | 76.16 | 73.56 |
| Phi3-7B | 71.43 | 76.68 | 73.96 |
| **GPT-3.5-turbo** | **71.76** | **77.00** | **74.29** |

**(a)** Impact of large language model.

| Models | Base | Novel | H |
|---|---|---|---|
| FUYU-8B | 70.98 | 75.45 | 73.14 |
| BLIP-2 | 71.95 | 76.52 | 73.16 |
| Qwen-VL-Chat-9.6B | 71.23 | **77.08** | 74.03 |
| LLaVA-Llama-3-8B | **73.17** | 75.24 | 74.19 |
| **MiniCPM-V-2-2.8B** | 71.76 | 77.00 | **74.29** |

**(b)** Impact of large multimodal model.

**Table 2:** Effect of LLM and LMM choice. We obtain best results with GPT-3.5-turbo as the LLM optimizer, and MiniCPM-V-2.0 for generating image descriptions for the Prompt Optimization Prompt.

in the prompt. This understanding can guide us in identifying crucial words that enhance prompt effectiveness.

**Effect of LLM and LMM choice.** In Table 2, we analyze the effect of the choice of the LLM optimizer and LMM for generating image descriptions as Prompt Optimization Prompt inputs. We evaluate various models and report their average performance across 11 datasets. Using GPT-3.5-turbo as the optimizer results in better prompt generation, especially improving performance on novel classes. The training speed of our IPO is heavily influenced by the computational efficiency of LLMs used. Since PaLM and GPT-3.5 are not open-source, we rely on their respective APIs to generate prompts. Consequently, our training speed depends on the API call latency and the computational complexity of these models. Alternatively, Phi2 and Phi3 are open-source, allowing us to generate prompts directly using their weights. Therefore, for researchers and practitioners seeking faster and more cost-effective training, we recommend utilizing open-source large models for prompt generation. When comparing different LMMs, we found that LLaVA-Llama-3-8B and MiniCPM-V-2.0 perform

| Dataset | Best Prompt |
|---|---|
| ImageNet | Take a high-quality photo of a <CLASS>. |
| Caltech101 | Categorize the <CLASS> shown in the image. |
| OxfordPets | Take a well-composed photo of a <CLASS> with optimal lighting, focus, and minimal distractions. Capture the pet's unique characteristics, including expression and posture, to ensure a clear and distinct image. |
| StanfordCars | Describe the distinguishing characteristics of the <CLASS> in the image. |
| Flowers102 | Identify the unique visual features of the <CLASS> flower accurately. |
| Food101 | Identify the primary ingredient in the <CLASS> and describe its texture, color, and presentation. |
| FGVCAircraft | Capture a comprehensive range of well-lit, high-resolution images of an <CLASS> from various angles, meticulously showcasing its specific design features with perfect clarity and precision for unparalleled accuracy in aircraft. |
| SUN397 | A photo of a <CLASS>, a type of large-scale scene. |
| DTD | Classify the intricate <CLASS> texture. |
| EuroSAT | Analyze the <CLASS> vehicles in the satellite image with state-of-the-art algorithms for precise classification and optimal efficiency. |
| UCF101 | Capture a high-quality, well-lit image of a person flawlessly demonstrating the <CLASS> action with impeccable visual representation to achieve unmatched. |

**Table 3:** Interpretable prompts generated by our method for each dataset in 1-shot scenarios.

almost identically on average. However, MiniCPM-V-2.0 shows better performance on novel classes. Before using LLMs, we first use LMMs to generate image descriptions, which are then used as inputs for the LLMs. In terms of computation cost, MiniCPM-V-2.0, with its smaller parameter size, generates descriptions more quickly. Therefore, we recommend this lightweight LMM as the image description generator for more efficient processing. In summary, the text comprehension capabilities of LLMs are crucial for determining the quality of the optimized prompts.

**Interpretable prompts generated per dataset.** Table 3 showcases the diverse prompts generated by our optimizer for each dataset. Our method enhances model accuracy by concentrating on the most relevant attributes, such as those in Flowers102 [52] and Food101 [17]. Consequently, it delivers high-quality text prompts that improve vision-language models. We encourage future researchers to leverage these interpretable prompts in their own downstream tasks.

**Benefit of image description.** In Table 4, we assess the impact of incorporating image descriptions into the Prompt Optimization Prompt on the performance of the optimizer. The inclusion of image descriptions enhances the model's performance. This improvement suggests that IPO, when generating new prompts, can effectively integrate information from the images themselves. As a result, the optimizer is able to produce prompts that are more specific to the data, thereby increasing the relevance and accuracy of the generated content. This highlights the importance of multimodal inputs in optimizing the prompting abilities of vision-language models.

|  | Base | New | H |
|---|---|---|---|
| CLIP | 69.34 | 74.22 | 71.70 |
| w/o LMM | 71.12 | 76.03 | 73.49 |
| w/ LMM | **71.76** | **77.00** | **74.29** |

**Table 4:** Benefit of image description.

**Comparison with knowledge bank-based prompt learning methods.** Our IPO leverages LLMs to optimize prompts, which conceptually aligns with previous bank-based prompt learning approaches. To assess its relative effectiveness, we compared IPO to traditional bank-based methods, specifically using L2P [57], which learns to dynamically prompt (L2P) a pre-trained model to learn tasks sequentially under different task transitions. We apply L2P within the visual prompt tuning (VPT) [12] framework, which also learns prompts in the visual space and applies them to few-shot vision-language model tasks. Similar to our IPO, VPT + L2P learns prompts within the visual space and applies them to few-shot VLM tasks. In this setup, VPT + L2P trains a prompt bank, allowing test samples to query the bank for suitable prompts during testing. The table 5 presents a comparison of VPT + L2P and our method across 11 datasets in the 16-shot setting. While L2P contributes to enhanced VPT performance, confirming

| Model | Base | Novel | H |
|---|---|---|---|
| VPT [12] | 69.34 | 74.22 | 71.70 |
| VPT [12] + L2P [57] | 71.12 | 76.03 | 73.49 |
| VPT [12] + IPO | 71.76 | 77.00 | 74.29 |

**Table 5:** Comparison with knowledge bank-based prompt learning methods.

| ViT-B/16 | Base | Novel | H |
|---|---|---|---|
| CLIP | 69.34 | 74.22 | 71.70 |
| CoOp | 72.08 | 66.71 | 69.29 |
| CoCoOp | 72.85 | 72.17 | 72.51 |
| MaPLe | 70.85 | 71.57 | 71.21 |
| PromptSRC | **73.38** | 71.47 | 72.41 |
| CoPrompt | 70.44 | 70.11 | 70.27 |
| **IPO** | 71.76 | **77.00** | **74.29** |

**(a)** Average over 11 datasets.

| ViT-B/16 | Base | Novel | H |
|---|---|---|---|
| CLIP | 72.43 | 68.14 | 70.22 |
| CoOp | 73.20 | 67.43 | 70.20 |
| CoCoOp | 73.90 | 69.07 | 71.40 |
| MaPLe | 74.03 | 68.73 | 71.28 |
| PromptSRC | 73.27 | 68.87 | 71.00 |
| CoPrompt | 73.97 | **70.87** | **72.39** |
| **IPO** | **74.09** | 69.17 | 71.54 |

**(b)** ImageNet

| ViT-B/16 | Base | Novel | H |
|---|---|---|---|
| CLIP | 96.84 | 94.00 | 95.40 |
| CoOp | 90.63 | 85.20 | 87.83 |
| CoCoOp | 96.37 | 93.13 | 94.72 |
| MaPLe | 96.40 | 94.10 | 95.24 |
| PromptSRC | 97.30 | **95.57** | 96.43 |
| CoPrompt | **97.60** | **95.57** | **96.57** |
| **IPO** | 96.53 | 95.39 | 95.95 |

**(c)** Caltech101

| ViT-B/16 | Base | Novel | H |
|---|---|---|---|
| CLIP | 91.17 | 97.26 | 94.12 |
| CoOp | 93.73 | 96.23 | 94.96 |
| CoCoOp | 93.47 | 96.27 | 94.85 |
| MaPLe | 90.83 | 96.00 | 93.34 |
| PromptSRC | 93.73 | 97.33 | 95.50 |
| CoPrompt | 92.37 | 96.37 | 94.33 |
| **IPO** | **94.48** | **97.93** | **96.43** |

**(d)** OxfordPets

| ViT-B/16 | Base | Novel | H |
|---|---|---|---|
| CLIP | 63.37 | 74.89 | 68.65 |
| CoOp | 61.80 | 68.33 | 64.90 |
| CoCoOp | 65.27 | 73.73 | 69.24 |
| MaPLe | 66.00 | 73.67 | 69.62 |
| PromptSRC | **67.93** | 73.73 | **70.71** |
| CoPrompt | 64.17 | 71.50 | 67.64 |
| **IPO** | 63.83 | **75.45** | 69.16 |

**(e)** StanfordCars

| ViT-B/16 | Base | Novel | H |
|---|---|---|---|
| CLIP | 72.08 | 77.80 | 74.83 |
| CoOp | 83.97 | 67.10 | 74.59 |
| CoCoOp | 75.57 | 77.00 | 76.28 |
| MaPLe | 77.10 | 76.97 | 77.03 |
| PromptSRC | **85.57** | 74.83 | **79.84** |
| CoPrompt | 72.90 | 72.93 | 72.91 |
| **IPO** | 74.17 | **79.65** | 76.81 |

**(f)** Flowers102

| ViT-B/16 | Base | Novel | H |
|---|---|---|---|
| CLIP | **90.10** | 91.22 | 90.66 |
| CoOp | 87.90 | 88.03 | 87.96 |
| CoCoOp | 88.73 | 89.60 | 89.16 |
| MaPLe | 89.13 | 90.67 | 89.89 |
| PromptSRC | 88.30 | 91.03 | 89.64 |
| CoPrompt | 88.40 | 90.60 | 89.49 |
| **IPO** | 89.78 | **91.59** | **90.67** |

**(g)** Food101

| ViT-B/16 | Base | Novel | H |
|---|---|---|---|
| CLIP | 27.19 | 36.29 | 31.09 |
| CoOp | 27.77 | 27.60 | 27.68 |
| CoCoOp | 29.77 | 31.23 | 30.48 |
| MaPLe | 28.33 | 29.00 | 28.66 |
| PromptSRC | 10.93 | 6.73 | 8.33 |
| CoPrompt | 10.10 | 4.87 | 6.57 |
| **IPO** | **31.43** | **36.32** | **33.70** |

**(h)** FGVCAircraft

| ViT-B/16 | Base | Novel | H |
|---|---|---|---|
| CLIP | 69.36 | 75.35 | 72.23 |
| CoOp | 71.47 | 72.47 | 71.97 |
| CoCoOp | 73.67 | 75.50 | 74.57 |
| MaPLe | 74.33 | 76.37 | 75.34 |
| PromptSRC | 75.60 | 77.07 | 76.33 |
| CoPrompt | **76.37** | **78.77** | **77.55** |
| **IPO** | 72.25 | 77.53 | 74.80 |

**(i)** SUN397

| ViT-B/16 | Base | Novel | H |
|---|---|---|---|
| CLIP | 53.24 | 59.90 | 56.37 |
| CoOp | 60.80 | 47.53 | 53.35 |
| CoCoOp | 58.70 | 52.70 | 55.54 |
| MaPLe | 58.20 | 54.17 | 56.11 |
| PromptSRC | **63.17** | 55.60 | 59.14 |
| CoPrompt | 62.77 | 60.40 | **61.56** |
| **IPO** | 55.45 | **62.47** | 58.75 |

**(j)** DTD

| ViT-B/16 | Base | Novel | H |
|---|---|---|---|
| CLIP | 56.48 | 64.05 | 60.03 |
| CoOp | 69.13 | 50.33 | 58.25 |
| CoCoOp | **71.13** | 62.87 | 66.75 |
| MaPLe | 50.20 | 51.20 | 50.70 |
| PromptSRC | 73.27 | 67.00 | 70.00 |
| CoPrompt | 59.27 | 51.60 | 55.17 |
| **IPO** | 64.97 | **82.13** | **72.54** |

**(k)** EuroSAT

| ViT-B/16 | Base | Novel | H |
|---|---|---|---|
| CLIP | 70.53 | 77.50 | 73.85 |
| CoOp | 72.50 | 63.57 | 67.74 |
| CoCoOp | 74.73 | 72.80 | 73.75 |
| MaPLe | 74.83 | 76.43 | 75.62 |
| PromptSRC | **78.13** | 78.37 | **78.25** |
| CoPrompt | 76.93 | 77.73 | 77.33 |
| **IPO** | 72.43 | **79.35** | 75.73 |

**(l)** UCF101

**Table 6:** Comparison with existing state-of-the-art methods for base-to-novel generalization using 1-shot learning. Except for CLIP, the results for other methods are based on our reimplementation of their official code. Our proposed IPO exhibits robust generalization capability and achieves significant improvements on novel classes across 11 datasets.

L2P's efficacy in the VLM domain, our IPO method still demonstrates superior results compared to VPT + L2P. This comparison, which we will include in the revised manuscript, reinforces that IPO 's effectiveness is not merely due to a memory retrieval mechanism but also benefits from prompt optimization through LLMs.

**Comparison with state-of-the-art.** Table 6 shows the comparative experiments of IPO against the state-of-the-art across 11 datasets in a 1-shot setting. Our method excels in the novel classes, surpassing the second-best performer, the original CLIP [30], by 2.78% in average performance. Other methods do not perform as well as CLIP in novel classes, indicating overfitting to base classes. For instance, CoCoOP [11] performs better in base classes but falls behind the original CLIP by 2.05% in novel classes. In particular, on the most challenging FGVCAircraft [53], and EuroSAT [54] dataset, the current state-of-the-art model, CoPrompt [32], performs poorly. This is because methods based on prefix-tuning, like CoPrompt [32], PromptSRC [28], require substantial amounts of data for training to achieve adequate generalization. Consequently, on more challenging datasets, when data is scarce, it becomes difficult to fine-

| | Base | Novel | H |
|---|---|---|---|
| CLIP [30] | 69.34 | 74.22 | 71.70 |
| CoOP [8] | 82.69 | 63.22 | 71.66 |
| CoCoOp [11] | 80.47 | 71.69 | 75.83 |
| MaPLe [13] | 82.28 | 75.14 | 78.55 |
| PromptSRC [28] | **84.26** | 76.10 | 79.97 |
| CoPrompt [32] | 84.00 | 77.23 | **80.48** |
| **IPO** (1-shot) | 71.76 | 77.00 | 74.29 |
| **IPO** (16-shot) | 79.92 | **80.51** | 80.21 |

**Table 7:** Comparison with gradient-based prompt learning methods for 16-shots across 11 datasets.

tune these prefixes effectively. In contrast, our model outperforms other methods, exceeding the second-best by 1.78% in harmonic mean. Additionally, in Table 7, we compared our method's performance with other traditional gradient-based prompt learning methods on a 16-shot setting across all datasets, where our approach consistently performs well in novel classes. Demonstrating that our approach can mitigate overfitting and generalize better to novel classes.

## 6 Conclusion

In this paper, we presented a novel approach to prompt optimization for vision-language models, addressing the limitations of existing gradient-descent-based methods. By integrating large language models for dynamic text prompt generation and optimization, we introduced the IPO system. This system guides LLMs in crafting effective prompts while maintaining a record of past prompts and their performance metrics, offering valuable in-context information. Additionally, we incorporated large multimodal models to generate image descriptions, enhancing the synergy between textual and visual modalities. Our comprehensive evaluation across 11 datasets demonstrated that our method improves the initial accuracy of vision-language models compared to traditional gradient-descent-based prompt learning methods. Most notably, our approach significantly enhances the interpretability of the generated prompts. By leveraging the strengths of LLMs, IPO ensures that the prompts remain human-understandable, thereby facilitating better transparency and oversight for vision-language models. This improvement in interpretability is crucial, as it allows for more effective and trustworthy human-AI collaboration, making vision-language systems more reliable and accessible.

**Limitation**. Our IPO method is primarily designed for few-shot scenarios. However, when dealing with large domain-specific datasets, the need to generate extensive image descriptions, which can lead to substantial computational costs due to the large text inputs required for LLMs. Currently, our model uses an input length of approximately 5,000 tokens. When scaled to larger datasets, the input length may increase to around 50,000 tokens. Using GPT-4 with an 8k context length, the cost for our current input size (5,000 tokens) is approximately 0.15 dollars per input (0.03 dollars per 1,000 tokens). For the expanded input size of 50,000 tokens, the cost would rise to approximately 3.00 dollars per input. If we were to use GPT-4 with a 32k context length, the cost for the 50,000-token input would be approximately 3.00 dollars for the first 32,000 tokens and an additional 1.08 dollars for the remaining 18,000 tokens, totaling approximately 4.08 dollars per input. Since our IPO method requires 100 iterations during training, the costs would multiply accordingly when scaled to large inputs. In future work, we aim to investigate methods for LMM fine-tuning to enable the direct input of both images and text, thereby generating even more sample-specific prompts.

**Broader Impact.** This paper explores the use of an LLM as an optimizer for refining text prompts in vision-language models. We introduce a straightforward yet interpretable approach to prompt optimization, which holds potential for societal impact, particularly in vision-language tasks.

## Acknowledgment

This work is financially supported by the Inception Institute of Artificial Intelligence, the University of Amsterdam and the allowance Top consortia for Knowledge and Innovation (TKIs) from the Netherlands Ministry of Economic Affairs and Climate Policy.

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

**(a) Average over 11 datasets.**

| ViT-B/16 | Base | Novel | H |
|---|---|---|---|
| CoOp | 82.69 | 63.22 | 71.66 |
| CoCoOp | 80.47 | 71.69 | 75.83 |
| MaPLe | 82.28 | 75.14 | 78.55 |
| PromptSRC | **84.26** | 76.10 | 79.97 |
| **IPO** | 79.92 | **80.51** | **80.21** |

**(b) ImageNet**

| ViT-B/16 | Base | Novel | H |
|---|---|---|---|
| CoOp | 76.47 | 67.88 | 71.92 |
| CoCoOp | 75.98 | 70.43 | 73.10 |
| MaPLe | 76.66 | 70.54 | 73.47 |
| PromptSRC | 77.60 | 70.73 | 74.01 |
| **IPO** | **77.83** | **72.45** | **75.04** |

**(c) Caltech101**

| ViT-B/16 | Base | Novel | H |
|---|---|---|---|
| CoOp | 98.00 | 89.81 | 93.73 |
| CoCoOp | 97.96 | 93.81 | 95.84 |
| MaPLe | 97.74 | 94.36 | 96.02 |
| PromptSRC | **98.10** | 94.03 | 96.02 |
| **IPO** | 97.32 | **95.23** | **96.26** |

**(d) OxfordPets**

| ViT-B/16 | Base | Novel | H |
|---|---|---|---|
| CoOp | 93.67 | 95.29 | 94.47 |
| CoCoOp | 95.20 | 97.69 | 96.43 |
| MaPLe | **95.43** | 97.76 | 96.58 |
| PromptSRC | 95.33 | 97.30 | 96.30 |
| **IPO** | 95.21 | **98.23** | **96.70** |

**(e) StanfordCars**

| ViT-B/16 | Base | Novel | H |
|---|---|---|---|
| CoOp | 78.12 | 60.40 | 68.13 |
| CoCoOp | 70.49 | 73.59 | 72.01 |
| MaPLe | 72.94 | 74.00 | 73.47 |
| PromptSRC | **78.27** | 74.97 | **76.58** |
| **IPO** | 73.42 | **75.71** | 74.55 |

**(f) Flowers102**

| ViT-B/16 | Base | Novel | H |
|---|---|---|---|
| CoOp | 97.60 | 59.67 | 74.06 |
| CoCoOp | 94.87 | 71.75 | 81.71 |
| MaPLe | 95.92 | 72.46 | 82.56 |
| PromptSRC | **98.07** | 76.50 | 85.95 |
| **IPO** | 96.78 | **78.32** | **86.58** |

**(g) Food101**

| ViT-B/16 | Base | Novel | H |
|---|---|---|---|
| CoOp | 88.33 | 82.26 | 85.19 |
| CoCoOp | 90.70 | 91.29 | 90.99 |
| MaPLe | 90.71 | 92.05 | 91.38 |
| PromptSRC | 90.67 | 91.53 | 91.10 |
| **IPO** | **90.92** | **93.08** | **91.99** |

**(h) FGVCAircraft**

| ViT-B/16 | Base | Novel | H |
|---|---|---|---|
| CoOp | 40.44 | 22.30 | 28.75 |
| CoCoOp | 33.41 | 23.71 | 27.74 |
| MaPLe | 37.44 | 35.61 | 36.50 |
| PromptSRC | **42.73** | 37.87 | 40.15 |
| **IPO** | 41.21 | **41.42** | **41.31** |

**(i) SUN397**

| ViT-B/16 | Base | Novel | H |
|---|---|---|---|
| CoOp | 80.60 | 65.89 | 72.517 |
| CoCoOp | 79.74 | 76.86 | 78.27 |
| MaPLe | 80.82 | 78.70 | 79.75 |
| PromptSRC | **82.67** | 78.47 | 80.52 |
| **IPO** | 81.25 | **80.92** | **81.08** |

**(j) DTD**

| ViT-B/16 | Base | Novel | H |
|---|---|---|---|
| CoOp | 79.44 | 41.18 | 54.24 |
| CoCoOp | 77.01 | 56.00 | 64.85 |
| MaPLe | 80.36 | 59.18 | 68.16 |
| PromptSRC | **83.37** | 62.97 | 71.75 |
| **IPO** | 82.14 | **66.81** | **73.69** |

**(k) EuroSAT**

| ViT-B/16 | Base | Novel | H |
|---|---|---|---|
| CoOp | 92.19 | 54.74 | 68.69 |
| CoCoOp | 87.49 | 60.04 | 71.21 |
| MaPLe | 94.07 | 73.23 | 82.35 |
| PromptSRC | 92.90 | 73.90 | 82.32 |
| **IPO** | **94.25** | **80.11** | **86.61** |

**(l) UCF101**

| ViT-B/16 | Base | Novel | H |
|---|---|---|---|
| CoOp | 84.69 | 56.05 | 67.46 |
| CoCoOp | 82.33 | 73.45 | 77.64 |
| MaPLe | 83.00 | 78.66 | 80.77 |
| PromptSRC | **87.10** | 78.80 | 82.74 |
| **IPO** | 85.32 | **80.92** | **83.06** |

**Table 8:** Comparison with existing state-of-the-art methods for base-to-novel generalization using 16-shots learning. Our proposed IPO exhibits robust generalization capability and achieves significant improvements on novel classes across 11 datasets.

| | Source | Target | | | | | | | | | | |
|---|---|---|---|---|---|---|---|---|---|---|---|---|
| | ImageNet | Caltech101 | OxfordPets | StanfordCars | Flowers102 | Food101 | Aircraft | SUN397 | DTD | EuroSAT | UCF101 | Average |
| CoOp | **71.51** | 93.70 | 89.14 | 64.51 | 68.71 | 85.30 | 18.47 | 64.15 | 41.92 | 46.39 | 66.55 | 63.88 |
| CoCoOp | 71.02 | **94.43** | 90.14 | 65.32 | 71.88 | 86.06 | 22.94 | 67.36 | 45.73 | 45.37 | 68.21 | 65.74 |
| IPO | 72.15 | 94.34 | **90.96** | **66.10** | **72.75** | **86.75** | **25.14** | **67.97** | **47.01** | **48.56** | **69.23** | **67.36** |

**Table 9: Cross-dataset generalization.** Accuracy (%) evaluation for prompts learned from the source dataset. Our IPO consistently outperforms existing prompt learning methods.

# A   Experiments on 16-Shots

We report the 16-shot performance of our IPO method across 11 datasets, providing detailed results for the Base, Novel, and H metrics in Table 8. Our IPO method consistently outperforms all other approaches on the novel classes and the H metric, highlighting its effectiveness in reducing overfitting.

# B   Experiments on cross-dataset

We conducted a comprehensive cross-dataset experimental evaluation using the standard 16-shot setting to assess the performance of our IPO method. The results, presented in Table 9, demonstrate that IPO consistently outperforms previous gradient-based prompt learning approaches. By applying our task-agnostic, LLM-driven prompt optimization technique, IPO achieved superior results across various datasets, showcasing its robustness and generalizability. These findings highlight the effectiveness of IPO in adapting to diverse tasks and domains, further reinforcing its advantage over traditional gradient-based methods in few-shot learning scenarios.

| LLM | Params | LMM | Params Base | Novel | H | |
|---|---|---|---|---|---|---|
| GPT-3.5-turbo | 175B | MiniCPM-V-2 | 2.8B | 71.76 | 77.00 | 74.29 |
| GPT-4 | 175B | MiniCPM-V-2 | 2.8B | 72.67 | 77.62 | 75.06 |
| GPT-4-o | 175B | MiniCPM-V-2 | 2.8B | 72.91 | 78.13 | 75.42 |
| GPT-3.5-turbo | 175B | GPT-4o | 500B ˜ 1T | 72.78 | 77.92 | 75.26 |
| GPT-4 | 500B ˜ 1T | GPT-4o | 500B ˜ 1T | 72.93 | 78.01 | 75.38 |
| GPT-4-o | 500B ˜ 1T | GPT-4o | 500B ˜ 1T | 73.41 | 78.93 | 76.06 |

**Table 10:** Impact of large language model.

| LLM | Params | Base | Novel | H |
|---|---|---|---|---|
| CLIP | - | 69.34 | 74.22 | 71.70 |
| w/o LMM | - | 71.12 | 76.03 | 73.49 |
| w/MiniCPM-V-2 | 2.8B | 71.76 | 77.00 | 74.29 |
| w/GPT-4o | 500B ˜ 1T | 72.78 | 77.92 | 75.26 |

**Table 11:** Impact of large language model.

## C  Impact of LLM

As demonstrated in Table 10, upgrading the LLM capacity yielded a similarly positive impact on performance. To explore how a more advanced LLM, like GPT-4, could generate more effective prompts for our model, we conducted additional experiments with both GPT-4 and GPT-4o. Specifically, when we enhanced the LLM to GPT-4o and paired it with the GPT-4o LMM, we observed a significant overall increase in the H-score by 1.77% compared to the initial setup using GPT-3.5-turbo alongside MiniCPM-V-2. This improvement underscores the advantages of employing larger, more capable models, as they facilitate greater task generalization and more robust performance. The findings suggest that scaling up model capacity in both the LMM and LLM components can lead to substantial gains in prompt quality and adaptability across various tasks, indicating a promising direction for further enhancing our model's versatility and effectiveness.

## D  Experiments on Segmentation Task with IPO

In prompt-based vision tasks like segmentation and detection, the design of the text prompt plays a pivotal role. Our task-agnostic method, IPO, can be seamlessly integrated into various vision tasks to optimize text prompts. For example, as shown in Table 13, we applied IPO to pre-trained semantic segmentation models [58, 59], where the original prompt used was 'a photo of a [CLASS].' By utilizing GPT-4o as both the LLM and LMM, we generated more effective, context-specific text prompts tailored to the open-vocabulary semantic segmentation task, leading to significant performance improvements. These results highlight the potential of IPO to enhance text prompt design in segmentation tasks, showcasing its adaptability and value. We plan to extend our exploration of IPO to other vision tasks in future studies, aiming to further validate its effectiveness in optimizing prompt construction across a broader range of applications.

IPO with GPT-3.5 Turbo, indeed, does not show an improvement on the large-scale ImageNet. This is because ImageNet has a large number of classes and samples, which results in longer LLM input when generating descriptions for each sample. GPT-3.5 Turbo has limited performance in handling long-text inputs. The table 12 shows the results on ImageNet when IPO uses GPT-4o, which has superior long-text understanding compared to GPT-3.5 Turbo. We found that IPO using GPT-4o leads to better performance improvements over other methods as well as a considerable improvement over IPO with GPT-3.5 Turbo.

## E  Experiments on segmentation task with IPO

In other prompt-based vision tasks, such as segmentation and detection, the design of the text prompt is crucial. Our method, being task-agnostic, can be easily embedded into any vision task to optimize

| Model | Base | Novel | H |
|---|---|---|---|
| CLIP | 72.43 | 68.14 | 70.22 |
| CoOp | 73.20 | 67.43 | 70.20 |
| CoCoOp | 73.90 | 69.07 | 71.40 |
| MaPLe | 74.03 | 68.73 | 71.28 |
| CoPrompt | 73.97 | 70.87 | 72.39 |
| IPO w/ GPT-3.5 | 74.09 | 69.17 | 71.54 |
| IPO w/ GPT-4o | 76.14 | 72.13 | 74.09 |

**Table 12:** Performance on large-scale generic datasets.

| Methods | pAcc | mIoU (S) | mIoU (U) | hIoU |
|---|---|---|---|---|
| SPNet [60] | - | 78.0 | 15.6 | 26.1 |
| ZS3 [61] | - | 77.3 | 17.7 | 28.7 |
| CaGNet [62] | 80.7 | 78.4 | 26.6 | 39.7 |
| SIGN [63] | - | 75.4 | 28.9 | 41.7 |
| Joint [64] | - | 77.7 | 32.5 | 45.9 |
| Zegformer [65] | - | 86.4 | 63.6 | 73.3 |
| Zsseg [58] | 90.0 | 83.5 | 72.5 | 77.5 |
| ZegCLIP [59] | 94.6 | 91.9 | 77.8 | 84.3 |
| Zsseg + IPO | 91.2 | 84.7 | 73.2 | 78.6 |
| ZegCLIP + IPO | 95.3 | 92.7 | 78.7 | 85.1 |

**Table 13:** Experiments on segmentation tasks.

the text prompt. For instance, as shown in Table 13, we incorporated IPO into pre-trained semantic segmentation models [58, 59], where the original text prompt was "a photo of a [CLASS]." Using GPT-4o as the LLM and LMM, we crafted more effective text prompts specifically suited to the open-vocabulary semantic segmentation task, leading to enhanced performance and demonstrating the value of IPO in optimizing text prompts for this application. We intend to further investigate the use of IPO in other vision tasks in future work.

# F   Effect of batch size

The Table 14 compares the performance of GPT-3.5 turbo and GPT-4o across different batch sizes. We observed that as the batch size increases to 128, GPT-3.5 turbo's performance begins to decline due to its limited capacity for handling longer input texts effectively. In contrast, GPT-4o maintains strong performance even at larger batch sizes. However, using extremely large batch sizes with GPT-4o becomes cost-prohibitive. Therefore, we selected a batch size of 128 for our experiments. Although

| Model | Batch size | Base | Novel | H |
|---|---|---|---|---|
| IPO w/ GPT-3.5 | 4 | 73.11 | 68.08 | 70.51 |
| IPO w/ GPT-4o | 4 | 74.32 | 67.98 | 70.55 |
| IPO w/ GPT-3.5 | 16 | 73.42 | 68.43 | 70.82 |
| IPO w/ GPT-4o | 16 | 74.94 | 70.75 | 72.78 |
| IPO w/ GPT-3.5 | 32 | 73.79 | 68.72 | 71.16 |
| IPO w/ GPT-4o | 32 | 75.01 | 70.93 | 72.91 |
| IPO w/ GPT-3.5 | 64 | 74.09 | 69.17 | 71.54 |
| IPO w/ GPT-4o | 64 | 75.34 | 71.23 | 73.45 |
| IPO w/ GPT-3.5 | 128 | 73.67 | 68.07 | 70.75 |
| IPO w/ GPT-4o | 128 | 76.14 | 72.13 | 74.09 |
| IPO w/ GPT-3.5 | 256 | 73.11 | 67.81 | 70.36 |
| IPO w/ GPT-4o | 256 | 76.81 | 72.73 | 74.71 |

**Table 14:** Effect of batch size.

| History length | Base | Novel | H |
|---|---|---|---|
| n = 0 | 69.15 | 75.20 | 72.04 |
| n = 1 | 70.25 | 75.43 | 72.74 |
| n = 5 | 70.95 | 76.21 | 73.49 |
| n = 10 | 71.23 | 76.41 | 73.72 |
| n = 20 | 71.76 | 77.00 | 74.29 |
| n = 50 | 71.81 | 76.81 | 74.23 |
| n = 100 | 72.02 | 76.81 | 74.33 |

**Table 15:** Impact of prompt history length.

even larger batch sizes could potentially improve performance further, the cost considerations become a critical factor.

## G    Impact of prompt history length

We evaluated the effect of varying prompt history lengths on model performance, as shown in Table 15. Our findings indicate that without prompt history, performance declines due to the absence of contextual information, making it challenging for the LLM to converge. As the history length increases, performance progressively improves, with convergence observed at n=20. Although using n=100 yields the highest average performance, the extended input length significantly raises API costs. As a result, we selected n=20 for our IPO, balancing performance gains with cost efficiency.

| Model | Base | Novel | H |
|---|---|---|---|
| LFA [66] | 83.62 | 74.56 | 78.83 |
| IPO | 79.92 | 80.51 | 80.21 |

| Model | 1-shot | 16-shots |
|---|---|---|
| PLOT [67] | 65.45 | 76.20 |
| IPO | 74.29 | 80.21 |

**(a)** Comparison with LFA across 11 datasets in 16-shot scenarios.

**(b)** Comparison with PLOT on average accuracy across 11 datasets in 1-shot and 16-shot scenarios.

**Table 16:** Comparison with recent prompt learning methods

## H    Comparison with recent prompt learning methods

We conducted a comparative evaluation with LFA [66] and PLOT [67] under the same experimental conditions, as shown in Tables 16. Our IPO method consistently outperforms both LFA and PLOT across the benchmarks.

## I    Detailed Prompt Optimization Prompt

In Figure 3, 4, 5 and 6, 7, 8, we show detailed inputs and outputs of different training steps in our Prompt Optimization Prompt. We observed that each optimized prompt is unique, and both loss and accuracy exhibit a downward trend.

## J    Loss and accuracy curve.

To demonstrate that IPO can indeed serve as optimizer for prompt learning in vision-language models, we present the optimization process's loss and accuracy on ImageNet in Figure 9. As the training steps increase, the loss consistently decreases, and the accuracy gradually improves, proving the optimization capability of IPO. Additionally, IPO not only allows for interpretable prompt generation but also reduces the risk of overfitting during training.

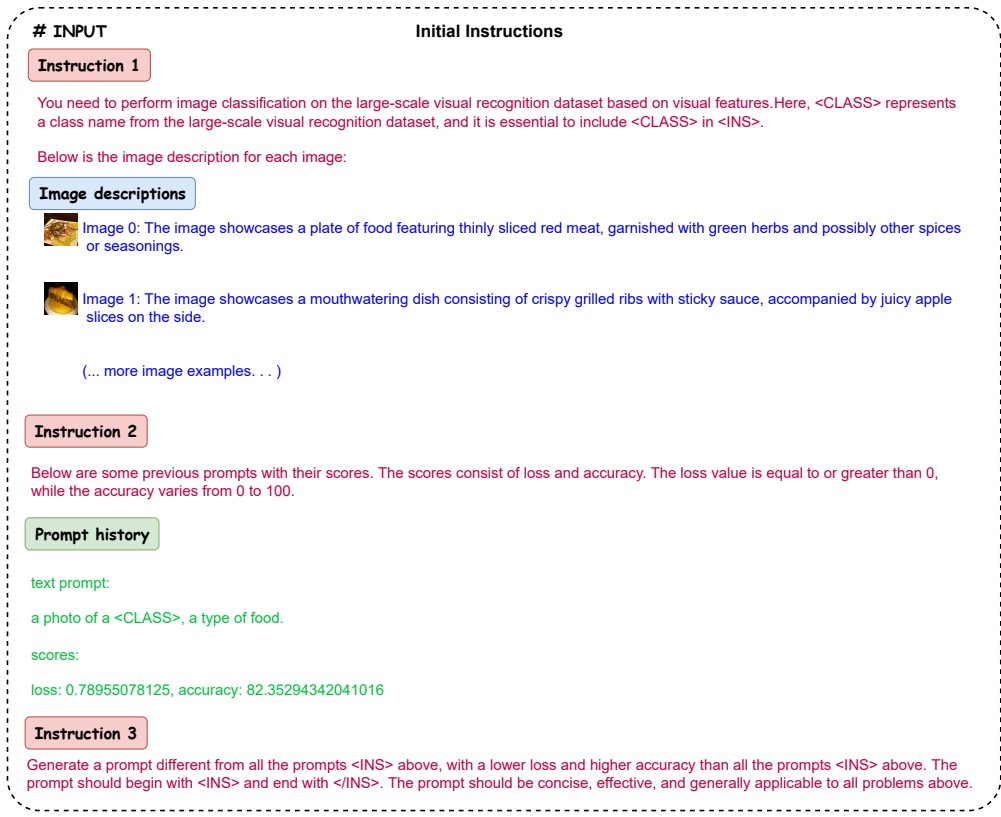

**Figure 3:** An example of our Prompt Optimization Prompt with input with initial instruction on the Food101 [17] dataset.

## K  More generated prompts

In Table 17, we provide detailed prompts for each dataset in 16-shot format. We encourage future researchers to utilize these interpretable prompts in their own downstream tasks.

## L  Image description with LMM

In Table 18, we provide descriptions of some training samples generated using Mini-CPM-V-2.0 on each dataset, which serve as input for image information in our Prompt Optimization Prompt. Note that we did not use this image information in the 16-shot setup due to the context length limitations of the language model.

## M  Prompt Optimization Prompt design

Our Prompt Optimization Prompt is a crucial component of our optimizer, serving to enhance the performance of the vision-language model by optimizing the prompts through the *prompt* LLM. Figure 2 displays an example of our Prompt Optimization Prompt. Initially, the instruction in the first segment of the Prompt Optimization Prompt defines the role of the LLM. It introduces two essential tokens,  and <CLASS>, which represent the prompt and category, respectively. The primary function of this instruction is to inform the LLM of its role and the contents of the Prompt Optimization Prompt, enabling a more effective understanding of the Prompt Optimization Prompt. Note that in this section, our model does not involve image information, hence the Prompt Optimization Prompt here lacks statements like "*Here is a description of some features of the flowers*

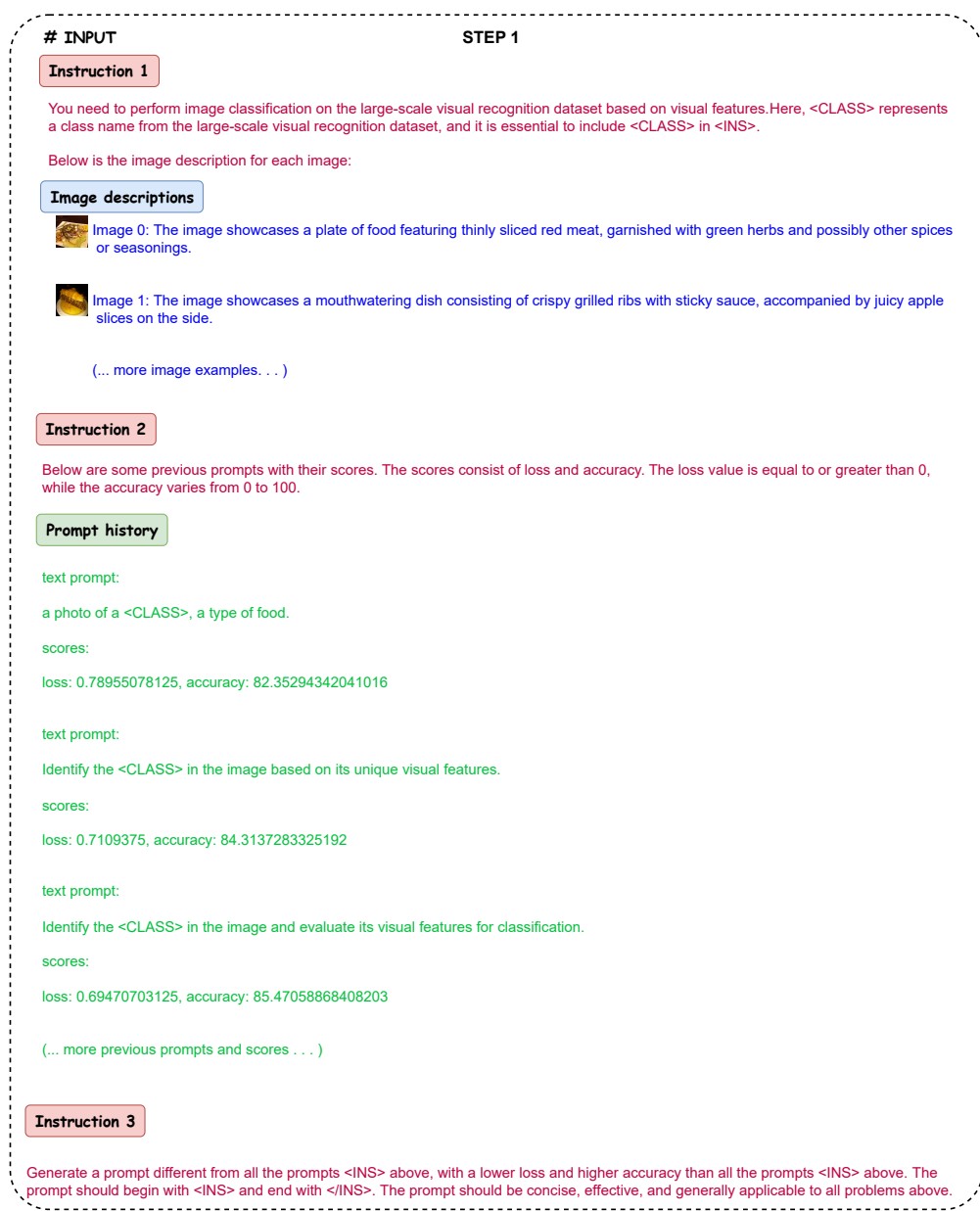

**# INPUT                                                        STEP 1**

**Instruction 1**

You need to perform image classification on the large-scale visual recognition dataset based on visual features.Here, <CLASS> represents a class name from the large-scale visual recognition dataset, and it is essential to include <CLASS> in .

Below is the image description for each image:

**Image descriptions**

Image 0: The image showcases a plate of food featuring thinly sliced red meat, garnished with green herbs and possibly other spices or seasonings.

Image 1: The image showcases a mouthwatering dish consisting of crispy grilled ribs with sticky sauce, accompanied by juicy apple slices on the side.

(... more image examples. . . )

**Instruction 2**

Below are some previous prompts with their scores. The scores consist of loss and accuracy. The loss value is equal to or greater than 0, while the accuracy varies from 0 to 100.

**Prompt history**

text prompt:

a photo of a <CLASS>, a type of food.

scores:

loss: 0.78955078125, accuracy: 82.35294342041016

text prompt:

Identify the <CLASS> in the image based on its unique visual features.

scores:

loss: 0.7109375, accuracy: 84.3137283325192

text prompt:

Identify the <CLASS> in the image and evaluate its visual features for classification.

scores:

loss: 0.69470703125, accuracy: 85.47058868408203

(... more previous prompts and scores . . . )

**Instruction 3**

Generate a prompt different from all the prompts  above, with a lower loss and higher accuracy than all the prompts  above. The prompt should begin with  and end with . The prompt should be concise, effective, and generally applicable to all problems above.

**Figure 4:** An example of our Prompt Optimization Prompt with input at step 1 on the Food101 [17] dataset.

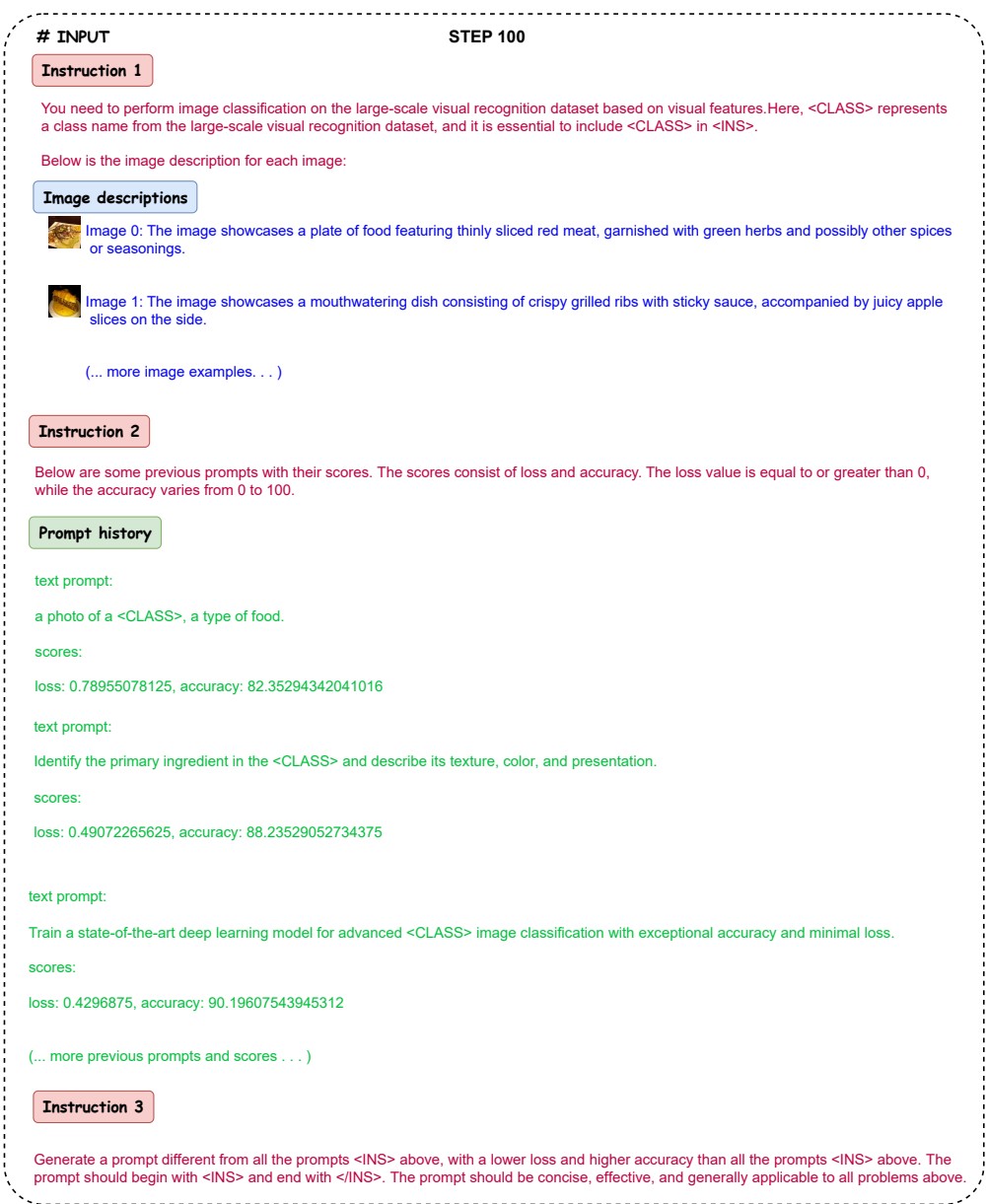

**# INPUT**                                    **STEP 100**

**Instruction 1**

You need to perform image classification on the large-scale visual recognition dataset based on visual features.Here, <CLASS> represents a class name from the large-scale visual recognition dataset, and it is essential to include <CLASS> in .

Below is the image description for each image:

**Image descriptions**

Image 0: The image showcases a plate of food featuring thinly sliced red meat, garnished with green herbs and possibly other spices or seasonings.

Image 1: The image showcases a mouthwatering dish consisting of crispy grilled ribs with sticky sauce, accompanied by juicy apple slices on the side.

(... more image examples. . . )

**Instruction 2**

Below are some previous prompts with their scores. The scores consist of loss and accuracy. The loss value is equal to or greater than 0, while the accuracy varies from 0 to 100.

**Prompt history**

text prompt:

a photo of a <CLASS>, a type of food.

scores:

loss: 0.78955078125, accuracy: 82.35294342041016

text prompt:

Identify the primary ingredient in the <CLASS> and describe its texture, color, and presentation.

scores:

loss: 0.49072265625, accuracy: 88.23529052734375

text prompt:

Train a state-of-the-art deep learning model for advanced <CLASS> image classification with exceptional accuracy and minimal loss.

scores:

loss: 0.4296875, accuracy: 90.19607543945312

(... more previous prompts and scores . . . )

**Instruction 3**

Generate a prompt different from all the prompts  above, with a lower loss and higher accuracy than all the prompts  above. The prompt should begin with  and end with . The prompt should be concise, effective, and generally applicable to all problems above.

**Figure 5:** An example of our Prompt Optimization Prompt with input step 100 on the Food101 [17] dataset.

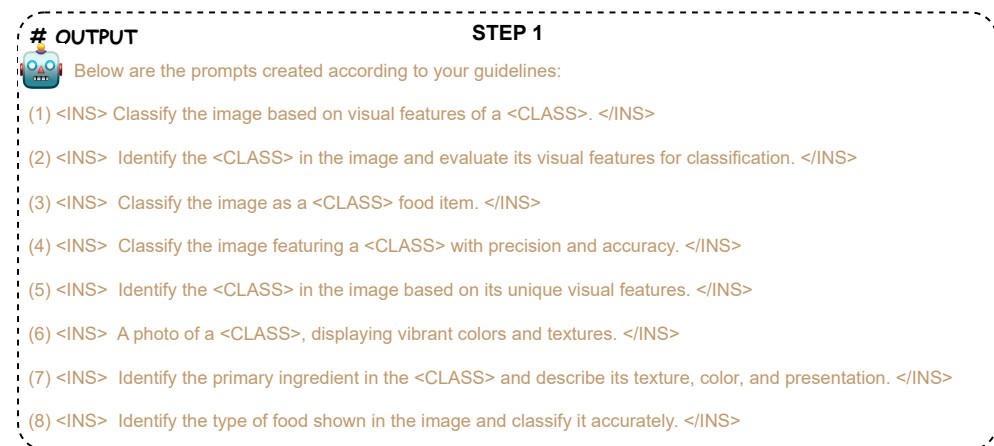

**Figure 6:** An example of our Prompt Optimization Prompt with output at step 1 on the Food101 [17] dataset.

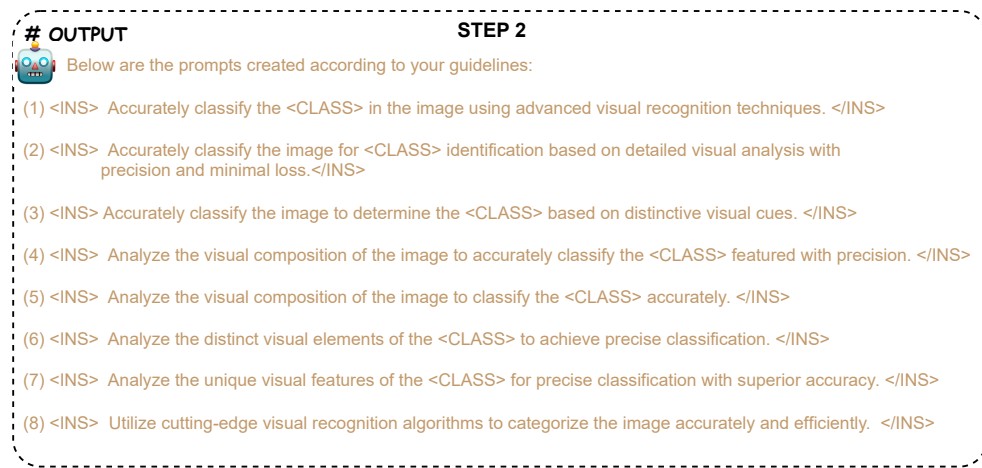

**Figure 7:** An example of our Prompt Optimization Prompt with output at step 2 on the Food101 [17] dataset.

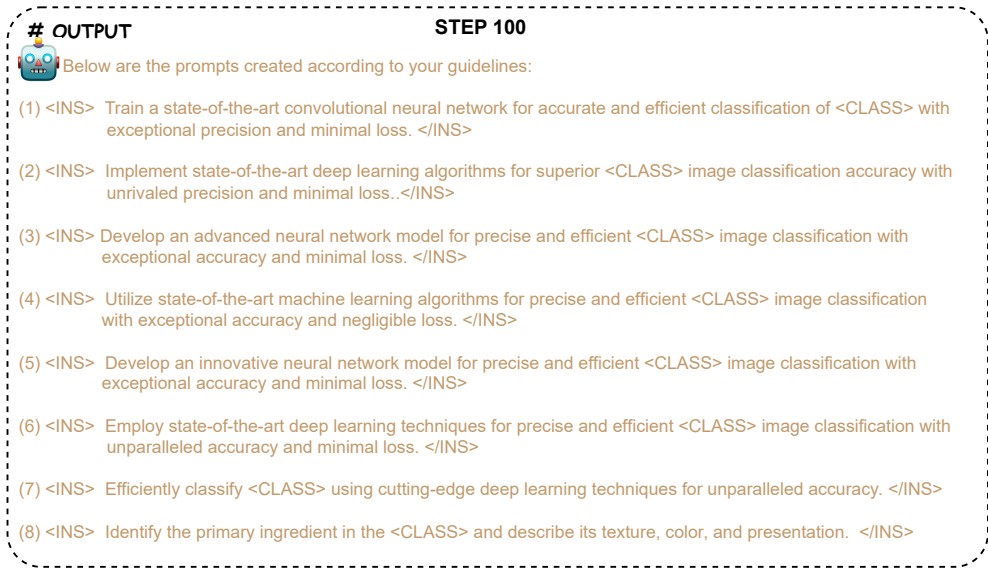

**Figure 8:** An example of our Prompt Optimization Prompt with output step 99 on the Food101 [17] dataset.

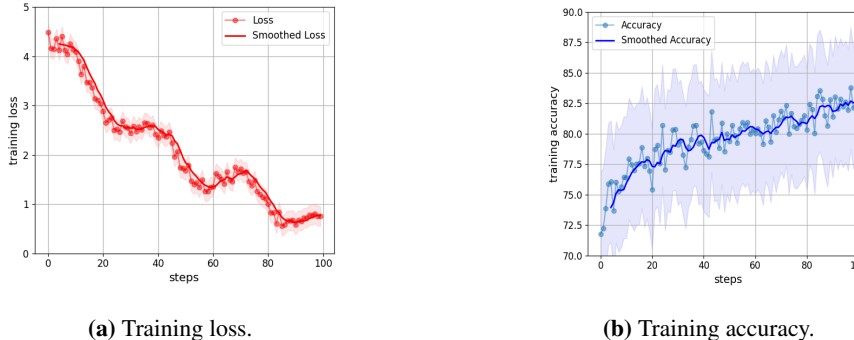

**(a)** Training loss.                    **(b)** Training accuracy.

**Figure 9:** Training loss and accuracy on ImageNet. Each dot represents the average loss and accuracy across up to 8 generated prompts in the single strip, with the shaded region indicating the standard deviation. Our findings demonstrate that IPO can effectively optimize prompt learning in vision-language models. Notably, the best performance was achieved at step 85.

*in the image*" and the subsequent description of the image. In the next section, we will discuss how to integrate image information into the Prompt Optimization Prompt.

The subsequent instruction concerns the history of the prompts and their associated scores, which include metrics such as loss and accuracy. Our task follows CoOP [8], focusing on base classes with few samples. We calculate the loss and accuracy for these base classes using the generated prompts. Initially, we inform the LLM that the following details are past prompts along with their scores, and we clarify the range of values for loss and accuracy. What follows are the historical prompts and scores. This history functions similarly to episodic memory, to prevent the generation of suboptimal prompts while providing in-context information that enhances the creation of better prompts. Additionally, as the history of past prompts grows, we only retain the top 20 prompts based on accuracy to avoid information overload. The selection of these top 20 prompts is determined by their accuracy scores. Moreover, we consistently include the prompt "`a photo of <CLASS>`" in our history at every step, as this is a frequently used and effective prompt within the CLIP framework [30].

| Dataset | Best Prompt |
|---|---|
| ImageNet | TLet's address problem, a photo of a <CLASS>. |
| Caltech101 | Take an awe-inspiring photograph of <CLASS> that beautifully captures its essence with exceptional clarity, vibrant colors, impeccable composition, and mesmerizing details. |
| OxfordPets | Capture a well-lit, high-resolution photo of the <CLASS> with optimal focus and minimal distractions. Ensure proper composition and framing to highlight its unique characteristics. Emphasize the pet's distinct traits by capturing its expression and posture accurately. |
| StanfordCars | Use a higher resolution camera to capture the vehicle photo <CLASS>.. |
| Flowers102 | Capture a high-resolution image of a <CLASS> in perfect lighting conditions, ensuring precise focus and a clutter-free background for maximum classification accuracy and to highlight the flower's distinctive features. |
| Food101 | Categorize the image depicting a delicious and appetizing <CLASS> with remarkable visual qualities. |
| FGVCAircraft | Capture a comprehensive set of high-resolution images of an <CLASS> from various angles, ensuring optimal lighting conditions and precise focus for unparalleled accuracy in aircraft model recognition. |
| SUN397 | A photo of a <CLASS>, a type of large-scale scene. |
| DTD | Capture an image depicting the distinct pattern of <CLASS>. |
| EuroSAT | Construct a state-of-the-art deep learning model on the Sentinel-2 satellite dataset for <CLASS> leveraging cutting-edge techniques including attention mechanisms, transfer learning, ensemble learning. |
| UCF101 | Capture a high-quality, well-lit image of a person flawlessly demonstrating the <CLASS> action, ensuring impeccable visual representation for unmatched results. |

**Table 17:** Interpretable prompts generated by our method for each dataset in 16-shot scenarios.

The final paragraph of the instruction specifies the ultimate goal of the task: to generate improved prompts based on the aforementioned instructions and the history of past prompts. The aim is to achieve lower loss and higher accuracy in this vision-language task, and it also outlines the format for the final generated prompt. These three components constitute the entire content of the Prompt Optimization Prompt, each playing a critical role for the LLM. It is important to note that in this Prompt Optimization Prompt, we do not consider the specific content of the images; we merely use images to calculate scores, and the details of the images are overlooked. In the following section, we will utilize a LMM to generate content from images and then incorporate it into the Prompt Optimization Prompt.

## N   More occlusion sensitivity analysis

In Tables 19-29, we present various prompts analyzed using occlusion sensitivity analysis across different models and datasets. We found that CLIP is particularly sensitive to photos. However, CoOP and CoCoOP exhibit severe overfitting on base classes, leading to poor performance on base classes when some tokens are removed, but improved performance on novel classes. In contrast, our optimized prompts show performance degradation to varying degrees when certain words or phrases are removed, indicating that the key words or phrases in our generated prompts are essential.

| Datasets | Images | Text description |
|---|---|---|
| Caltech101 |  | The cheetah's fur has spots and stripes, its eyes are black with white pupils. It is looking straight into the camera while standing against a clear blue sky background. |
| |  | The image prominently features a large white and red airplane with the Turkish flag logo, flying against a backdrop of blue sky dotted with clouds. |
| StanfordCars |  | The image features a black Audi TT parked outside the dealership, with its silver rims and elegant design. |
| |  | The car in the image is a 2016 Acura TSX, distinguished by its sleek design and distinctive tail lights. |
| Flowers102 |  | The flowers have a combination of pink and white colors with distinct leaves, creating an attractive appearance. |
| |  | The flowers in the image are large, have a yellow center with dark spots and petals that vary from pink to red. They appear vibrant due to their color contrast against natural backgrounds like leaves or rocks on trees behind them. |
| OxfordPets |  | The cat in the image is a light-colored, possibly an orange tabby with big yellow eyes and white whiskers. |
| |  | The image features a small white puppy with brown spots, which is likely to be an American Pit Bull Terrier breed. |
| Food101 |  | The food in the image is a delicious pastry with meat and cheese filling, covered by caramelized topping. |
| |  | The image features a colorful and diverse salad with various ingredients such as onions, corn kernels, lettuce leaves (cabbage), peanuts or pumpkin seeds. |
| FGVCAircraft |  | The aircraft in the image is a large commercial airplane, likely used for passenger transportation. It has an orange and white color scheme with prominent windows along its body to provide natural light inside during flights. |
| |  | The aircraft in the image is a Western Airlines Boeing 737, identified by its distinctive tail fin with "Western" written on it. It has various markings including numbers and letters that are part of their registration or identification system used for aviation purposes. |
| SUN397 |  | The image showcases a commercial airplane cabin with rows of blue and white chairs, each equipped with cup holders for passengers' convenience. |
| |  | The image captures the grandeur of a monument with an elaborate arch, surrounded by fountains and illuminated buildings at night. |
| DTD |  | The image showcases a close-up view of the texture on an object with distinct stripes, buttons and stitching details. |
| |  | The leaf has a yellowish background with darker spots and irregular black marks, indicating possible disease or damage. |
| ImageNet |  | The fish, characterized by its light yellow body and distinct blue fins with a black spot on the tail fin's edge. |
| |  | The image features a white chicken with red comb and orange legs, standing on wooden planks surrounded by straw or grass. |
| EuroSAT |  | The image features a black car with four-wheeled design, possibly indicating that it is an automobile. |
| |  | The image captures various cars, each with distinct features such as their color and shape. For instance, one car is red in color while another has a unique design that stands out from the rest of them. |
| UCF101 |  | A baby in a yellow shirt and blue jeans is actively crawling on the tiled floor, while an adult wearing red pants stands nearby. |
| |  | A female gymnast is skillfully performing on a balance beam, showcasing her athletic abilities as she navigates the challenging obstacle. |

**Table 18:** Image descriptions generated with Mini-CPM-V-2.0 [43].

| Models | Prompts | Base | Novel | H |
|---|---|---|---|---|
| CLIP [30] | a photo of a <CLASS>. | 72.43 | **68.14** | 70.22 |
| | <CLASS>. | 69.91 | 64.72 | 67.21 |
| | a <CLASS>. | 70.72 | 66.44 | 68.51 |
| | photo <CLASS>. | 68.15 | 62.32 | 65.10 |
| | of <CLASS>. | 67.84 | 64.93 | 66.35 |
| | a photo <CLASS>. | 69.91 | 64.15 | 66.91 |
| | photo of <CLASS>. | 71.24 | 65.67 | 68.34 |
| | of a <CLASS>. | 69.37 | 66.49 | 67.90 |
| | a photo of <CLASS>. | 70.99 | 64.26 | 67.46 |
| | photo of a <CLASS>. | **72.64** | 68.02 | **70.25** |
| CoOP [8] | Token 1, 2, 3, 4 | 73.20 | 67.43 | 70.20 |
| | None | 69.91 | 64.72 | 67.21 |
| | Token 1 | 66.39 | 62.88 | 64.59 |
| | Token 2 | 66.77 | 62.18 | 64.39 |
| | Token 3 | 74.06 | 68.61 | 71.23 |
| | Token 4 | 66.96 | 64.07 | 65.48 |
| | Token 1, 2 | 67.15 | 62.93 | 64.97 |
| | Token 3, 4 | **73.92** | **69.18** | **71.47** |
| | Token 1, 4 | 68.99 | 65.60 | 67.25 |
| | Token 1, 2, 3 | 73.15 | 66.54 | 69.69 |
| | Token 1, 2, 4 | 69.84 | 64.14 | 66.87 |
| | Token 1, 3, 4 | 73.52 | 68.64 | 71.00 |
| | Token 2, 3, 4 | 73.71 | 68.08 | 70.78 |
| CoCoOP [11] | Token 1, 2, 3, 4 | 73.90 | **69.07** | **71.40** |
| | None | 69.91 | 64.72 | 67.21 |
| | Token 1 | **74.01** | 67.85 | 70.80 |
| | Token 2 | 74.00 | 68.25 | 71.01 |
| | Token 3 | 73.52 | 68.58 | 70.96 |
| | Token 4 | 73.92 | 68.23 | 70.96 |
| | Token 1, 2 | 73.29 | 67.45 | 70.25 |
| | Token 3, 4 | 73.34 | 67.68 | 70.40 |
| | Token 1, 4 | 73.66 | 67.68 | 70.54 |
| | Token 1, 2, 3 | 71.45 | 65.57 | 68.38 |
| | Token 1, 2, 4 | 72.95 | 65.57 | 69.06 |
| | Token 1, 3, 4 | 72.52 | 66.87 | 69.58 |
| | Token 2, 3, 4 | 72.52 | 66.74 | 69.51 |
| IPO | Take a high-quality photo of a <CLASS>. | **74.09** | **69.17** | **71.54** |
| | <CLASS>. | 69.92 | 64.71 | 67.21 |
| | Take a photo of a <CLASS>. | 72.43 | 68.43 | 70.37 |
| | High-quality photo of a <CLASS>. | 73.64 | 68.24 | 70.84 |
| | Photo of a <CLASS>. | 72.65 | 68.08 | 70.29 |
| | Take a high-quality photo of a <CLASS>. | 74.08 | 69.12 | 71.51 |
| | Quality photo of a <CLASS>. | 73.69 | 68.23 | 70.85 |
| | High-quality <CLASS>. | 71.71 | 66.47 | 68.99 |
| | Take a photo of the <CLASS>. | 71.42 | 67.72 | 69.52 |
| | A photo of a <CLASS>. | 72.48 | 68.13 | 70.24 |

**Table 19:** Comparison of various prompts with occlusion sensitivity analysis across different models on the ImageNet dataset.

| Models | Prompts | Base | Novel | H |
|--------|---------|------|-------|---|
| CLIP [30] | a photo of a <CLASS>. | 96.84 | 94.00 | 95.40 |
| | <CLASS>. | 90.47 | 92.19 | 91.32 |
| | a <CLASS>. | 91.08 | 93.45 | 92.25 |
| | photo <CLASS>. | 89.67 | 90.62 | 90.14 |
| | of <CLASS>. | 91.38 | 92.72 | 92.05 |
| | a photo <CLASS>. | 92.02 | 92.69 | 92.35 |
| | photo of <CLASS>. | 91.93 | 93.32 | 92.62 |
| | of a <CLASS>. | 96.14 | **94.72** | 95.42 |
| | a photo of <CLASS>. | 91.32 | 94.45 | 92.86 |
| | photo of a <CLASS>. | **97.05** | 94.36 | **95.69** |
| CoOP [8] | Token 1, 2, 3, 4 | 90.63 | 85.20 | 87.83 |
| | None | 90.47 | 92.19 | 91.32 |
| | Token 1 | 89.29 | 90.41 | 89.85 |
| | Token 2 | 89.71 | 90.83 | 90.27 |
| | Token 3 | 87.18 | 92.76 | 89.88 |
| | Token 4 | 91.75 | 91.45 | 91.60 |
| | Token 1, 2 | 89.86 | 92.97 | 91.39 |
| | Token 3, 4 | 89.97 | 92.26 | 91.10 |
| | Token 1, 4 | 92.84 | 92.04 | 92.44 |
| | Token 1, 2, 3 | 88.13 | 92.26 | 90.15 |
| | Token 1, 2, 4 | **94.34** | **93.02** | **93.68** |
| | Token 1, 3, 4 | 88.56 | 92.81 | 90.64 |
| | Token 2, 3, 4 | 89.52 | 92.97 | 91.21 |
| CoCoOP [11] | Token 1, 2, 3, 4 | 96.37 | 93.13 | 94.72 |
| | None | 90.47 | 92.19 | 91.32 |
| | Token 1 | 96.36 | 93.38 | 94.85 |
| | Token 2 | 96.02 | 93.12 | 94.55 |
| | Token 3 | 96.97 | 93.38 | 95.14 |
| | Token 4 | 96.15 | 93.24 | 94.67 |
| | Token 1, 2 | 96.42 | **93.95** | 95.17 |
| | Token 3, 4 | **97.56** | **93.95** | **95.72** |
| | Token 1, 4 | 97.47 | 93.41 | 95.40 |
| | Token 1, 2, 3 | 95.54 | 92.87 | 94.19 |
| | Token 1, 2, 4 | 96.63 | 93.24 | 94.90 |
| | Token 1, 3, 4 | 97.41 | 93.24 | 95.28 |
| | Token 2, 3, 4 | 97.41 | 93.12 | 95.22 |
| IPO | Categorize the <CLASS> shown in the image. | 96.53 | 95.39 | 95.95 |
| | <CLASS>. | 90.45 | 92.16 | 91.30 |
| | Categorize <CLASS>. | 91.92 | 93.78 | 92.84 |
| | Categorize the <CLASS>. | 95.44 | 95.05 | 95.24 |
| | <CLASS> shown in the image. | 97.08 | 94.15 | 95.59 |
| | Shown in the image: <CLASS>. | 93.54 | 93.76 | 93.65 |
| | Categorize the <CLASS> in the image. | 96.64 | **95.74** | **96.19** |
| | The <CLASS> shown in the image. | 97.05 | 95.06 | 96.04 |
| | The <CLASS> in the image. | 93.54 | 94.32 | 93.93 |
| | Categorize the <CLASS> shown. | 95.37 | 95.09 | 95.23 |
| | Image of a <CLASS>. | **97.26** | 94.84 | 96.03 |

**Table 20:** Comparison of various prompts with occlusion sensitivity analysis across different models on the Caltech101 dataset.

| Models | Prompts | Base | Novel | H |
|---|---|---|---|---|
| CLIP [30] | a photo of a <CLASS>. | 89.42 | **96.81** | 92.97 |
| | <CLASS>. | 88.30 | 89.09 | 88.69 |
| | a <CLASS>. | 87.77 | 93.96 | 90.76 |
| | photo <CLASS>. | 84.69 | 90.66 | 87.57 |
| | of <CLASS>. | 86.98 | 88.53 | 87.75 |
| | a photo <CLASS>. | 87.19 | 91.16 | 89.13 |
| | photo of <CLASS>. | 88.62 | 90.38 | 89.49 |
| | of a <CLASS>. | 85.38 | 92.84 | 88.95 |
| | a photo of <CLASS>. | 88.94 | 91.39 | 90.15 |
| | photo of a <CLASS>. | **89.90** | 96.36 | **93.02** |
| CoOP [8] | Token 1, 2, 3, 4 | 93.73 | 96.23 | 94.96 |
| | None | 88.30 | 89.09 | 88.69 |
| | Token 1 | 81.34 | 86.80 | 83.98 |
| | Token 2 | 81.34 | 89.26 | 85.12 |
| | Token 3 | **93.83** | 94.80 | 94.31 |
| | Token 4 | 85.11 | 90.16 | 87.56 |
| | Token 1, 2 | 81.77 | 85.79 | 83.73 |
| | Token 3, 4 | 91.97 | 96.31 | 94.09 |
| | Token 1, 4 | 87.99 | 91.72 | 89.82 |
| | Token 1, 2, 3 | 93.62 | **97.37** | **95.46** |
| | Token 1, 2, 4 | 88.46 | 88.09 | 88.27 |
| | Token 1, 3, 4 | 91.92 | 97.32 | 94.54 |
| | Token 2, 3, 4 | 93.35 | **97.37** | 95.32 |
| CoCoOP [11] | Token 1, 2, 3, 4 | 93.47 | 96.27 | 94.85 |
| | None | 88.30 | 89.09 | 88.69 |
| | Token 1 | 94.26 | 95.64 | 94.94 |
| | Token 2 | 94.42 | 95.64 | 95.03 |
| | Token 3 | **94.84** | 96.14 | **95.49** |
| | Token 4 | 94.05 | 96.09 | 95.06 |
| | Token 1, 2 | 94.15 | 95.75 | 94.94 |
| | Token 3, 4 | 93.83 | **97.09** | 95.43 |
| | Token 1, 4 | 94.05 | 96.48 | 95.25 |
| | Token 1, 2, 3 | 93.67 | 95.30 | 94.48 |
| | Token 1, 2, 4 | 93.14 | 95.92 | 94.51 |
| | Token 1, 3, 4 | 93.35 | 96.59 | 94.94 |
| | Token 2, 3, 4 | 93.25 | 96.70 | 94.94 |
| IPO | Take a well-composed photo of a <CLASS> with optimal lighting, focus, and minimal distractions. Capture the pet's unique characteristics, including expression and posture, to ensure a clear and distinct image. | **94.48** | **97.93** | **96.43** |
| | <CLASS>. | 88.30 | 89.09 | 88.69 |
| | Take a photo of a <CLASS>. | 90.06 | 96.59 | 93.21 |
| | Well-composed photo of a <CLASS>. | 89.90 | 96.70 | 93.18 |
| | Photo of a <CLASS> with optimal lighting. | 89.37 | 96.81 | 92.94 |
| | <CLASS> with optimal lighting, focus, and minimal distractions. | 89.85 | 92.62 | 91.21 |
| | Capture the <CLASS>'s unique characteristics. | 87.77 | 89.60 | 88.68 |
| | Including expression and posture of the <CLASS>. | 86.98 | 88.98 | 87.97 |
| | Ensure a clear and distinct image of the <CLASS>. | 81.77 | 86.35 | 84.00 |
| | Unique characteristics of the <CLASS>. | 89.10 | 86.74 | 87.90 |
| | Capture the expression of the <CLASS>. | 88.20 | 90.04 | 89.11 |
| | Capture the posture of the <CLASS>. | 87.83 | 91.11 | 89.44 |
| | Expression of the <CLASS>. | 87.56 | 90.32 | 88.92 |
| | Posture of the <CLASS>. | 89.79 | 92.67 | 91.21 |
| | Focus and minimal distractions for a <CLASS>. | 90.27 | 96.31 | 93.19 |
| | Clear and distinct image of the <CLASS>. | 83.57 | 85.91 | 84.72 |

**Table 21:** Comparison of various prompts with occlusion sensitivity analysis across different models on the OxfordPets dataset.

| Models | Prompts | Base | Novel | H |
|---|---|---|---|---|
| CLIP [30] | a photo of a <CLASS>. | 63.37 | 74.89 | 68.65 |
| | <CLASS>. | 62.72 | 73.58 | 67.72 |
| | a <CLASS>. | 61.37 | 72.82 | 66.61 |
| | photo <CLASS>. | 61.79 | 73.78 | 67.25 |
| | of <CLASS>. | 62.04 | 73.98 | 67.49 |
| | a photo <CLASS>. | 62.57 | 73.98 | 67.80 |
| | photo of <CLASS>. | 63.09 | 74.75 | 68.43 |
| | of a <CLASS>. | 59.25 | 71.82 | 64.93 |
| | a photo of <CLASS>. | **63.82** | **75.41** | **69.13** |
| | photo of a <CLASS>. | 63.04 | 74.45 | 68.27 |
| CoOP [8] | Token 1, 2, 3, 4 | 61.80 | 68.33 | 64.90 |
| | None | 62.72 | 73.58 | **67.72** |
| | Token 1 | 61.42 | 72.05 | 66.31 |
| | Token 2 | 58.77 | 69.74 | 63.79 |
| | Token 3 | 62.37 | 68.31 | 65.21 |
| | Token 4 | 60.24 | 70.19 | 64.84 |
| | Token 1, 2 | 61.79 | 73.95 | 67.33 |
| | Token 3, 4 | 59.57 | 68.46 | 63.71 |
| | Token 1, 4 | 60.14 | 71.97 | 65.53 |
| | Token 1, 2, 3 | **63.37** | 69.82 | 66.44 |
| | Token 1, 2, 4 | 61.12 | **74.00** | 66.95 |
| | Token 1, 3, 4 | 60.79 | 70.29 | 65.20 |
| | Token 2, 3, 4 | 57.95 | 65.66 | 61.56 |
| CoCoOP [11] | Token 1, 2, 3, 4 | 65.27 | 73.73 | 69.24 |
| | None | 62.72 | 73.58 | 67.72 |
| | Token 1 | 65.22 | **75.27** | **69.89** |
| | Token 2 | 65.32 | 74.81 | 69.74 |
| | Token 3 | 65.09 | 74.05 | 69.28 |
| | Token 4 | **65.54** | 74.50 | 69.73 |
| | Token 1, 2 | 64.57 | 74.94 | 69.37 |
| | Token 3, 4 | 64.67 | 73.93 | 68.99 |
| | Token 1, 4 | 64.84 | 74.85 | 69.49 |
| | Token 1, 2, 3 | 64.07 | 74.52 | 68.90 |
| | Token 1, 2, 4 | 64.74 | 74.10 | 69.10 |
| | Token 1, 3, 4 | 63.17 | 74.18 | 68.23 |
| | Token 2, 3, 4 | 63.29 | 74.18 | 68.30 |
| IPO | Describe the distinguishing characteristics of the <CLASS> in the image. | 63.83 | **75.45** | **69.16** |
| | <CLASS>. | 62.72 | 73.58 | 67.72 |
| | Describe the <CLASS>. | 62.67 | 75.34 | 68.42 |
| | Distinguishing characteristics of the <CLASS>. | 63.52 | 74.40 | 68.53 |
| | Characteristics of the <CLASS> in the image. | 63.72 | 74.35 | 68.63 |
| | Describe the <CLASS> in the image. | 62.97 | 74.82 | 68.39 |
| | The <CLASS> in the image. | 63.97 | 74.94 | 69.02 |
| | Describe distinguishing characteristics of the <CLASS>. | 63.97 | 74.15 | 68.68 |
| | In the image, the <CLASS>. | 63.67 | 75.19 | 68.95 |
| | The <CLASS>'s distinguishing characteristics. | 61.34 | 72.39 | 66.41 |
| | The distinguishing characteristics of the <CLASS> in the image. | **64.07** | 74.72 | 68.99 |

**Table 22:** Comparison of various prompts with occlusion sensitivity analysis across different models on the StanfordCars dataset.

| Models | Prompts | Base | Novel | H |
|---|---|---|---|---|
| CLIP [30] | a photo of a <CLASS>. | **69.34** | 76.72 | **72.84** |
| | <CLASS>. | 61.75 | 75.83 | 68.07 |
| | a <CLASS>. | 63.04 | 75.02 | 68.51 |
| | photo <CLASS>. | 64.72 | 76.74 | 70.22 |
| | of <CLASS>. | 63.02 | 76.17 | 68.97 |
| | a photo <CLASS>. | 66.52 | **77.25** | 71.48 |
| | photo of <CLASS>. | 61.33 | 77.12 | 68.32 |
| | of a <CLASS>. | 63.37 | 75.91 | 69.08 |
| | a photo of <CLASS>. | 59.15 | **77.25** | 70.00 |
| | photo of a <CLASS>. | 69.16 | 76.72 | 72.74 |
| CoOP [8] | Token 1, 2, 3, 4 | 71.47 | 72.47 | 71.97 |
| | None | 61.75 | **75.83** | 68.07 |
| | Token 1 | 76.02 | 73.38 | **74.68** |
| | Token 2 | 70.07 | 74.26 | 72.10 |
| | Token 3 | 69.62 | 72.88 | 71.21 |
| | Token 4 | 70.85 | 67.75 | 69.27 |
| | Token 1, 2 | 75.04 | 74.23 | 74.63 |
| | Token 3, 4 | 70.13 | 66.75 | 68.40 |
| | Token 1, 4 | 75.92 | 68.81 | 72.20 |
| | Token 1, 2, 3 | 72.55 | 74.04 | 73.29 |
| | Token 1, 2, 4 | 74.82 | 70.52 | 72.61 |
| | Token 1, 3, 4 | **76.25** | 66.15 | 70.84 |
| | Token 2, 3, 4 | 70.91 | 67.93 | 69.39 |
| CoCoOP [11] | Token 1, 2, 3, 4 | 73.67 | 75.50 | 74.57 |
| | None | 61.75 | 75.83 | 68.07 |
| | Token 1 | 75.83 | 76.92 | 76.37 |
| | Token 2 | **76.75** | 76.98 | 76.86 |
| | Token 3 | 76.74 | 77.27 | 77.00 |
| | Token 4 | 76.51 | 77.64 | 77.07 |
| | Token 1, 2 | 76.02 | 77.23 | 76.62 |
| | Token 3, 4 | 76.07 | **78.75** | **77.39** |
| | Token 1, 4 | 76.49 | 78.22 | 77.35 |
| | Token 1, 2, 3 | 76.51 | 75.53 | 76.02 |
| | Token 1, 2, 4 | 76.04 | 77.92 | 76.97 |
| | Token 1, 3, 4 | 75.53 | 78.37 | 76.92 |
| | Token 2, 3, 4 | 75.25 | 77.71 | 76.46 |
| IPO | Identify the unique visual features of the <CLASS> flower accurately. | **74.17** | **79.65** | **76.81** |
| | <CLASS>. | 61.75 | 75.83 | 68.07 |
| | Identify <CLASS>. | 69.53 | 77.52 | 73.31 |
| | Identify the <CLASS>. | 68.88 | 76.35 | 72.42 |
| | Identify the unique <CLASS>. | 67.22 | 77.73 | 72.09 |
| | Identify unique visual features <CLASS>. | 63.28 | 77.21 | 69.55 |
| | Visual features of the <CLASS>. | 64.42 | 74.64 | 69.15 |
| | The unique visual features <CLASS> flower. | 66.42 | 76.67 | 71.18 |
| | Features of the <CLASS> flower accurately. | 72.03 | 77.82 | 74.81 |
| | Visual features of the <CLASS> flower. | 71.93 | 78.55 | 75.09 |
| | Identify the unique visual features of the <CLASS> accurately. | 65.67 | 77.35 | 71.03 |
| | Identify the unique visual features of the <CLASS> flower. | 73.82 | 79.11 | 76.37 |
| | The unique visual features of the <CLASS> flower accurately. | 72.67 | 78.72 | 75.57 |

**Table 23:** Comparison of various prompts with occlusion sensitivity analysis across different models on the Flowers102 dataset.

| Models | Prompts | Base | Novel | H |
|---|---|---|---|---|
| CLIP [30] | a photo of a <CLASS>. | 89.44 | **90.68** | **90.06** |
| | <CLASS>. | 89.25 | 89.44 | 89.34 |
| | a <CLASS>. | 88.48 | 89.12 | 88.80 |
| | photo <CLASS>. | 89.39 | 88.49 | 88.94 |
| | of <CLASS>. | **89.51** | 90.01 | 89.76 |
| | a photo <CLASS>. | 89.08 | 87.87 | 88.48 |
| | photo of <CLASS>. | 89.38 | 90.00 | 89.69 |
| | of a <CLASS>. | 89.31 | 90.17 | 89.74 |
| | a photo of <CLASS>. | 89.32 | 88.15 | 88.73 |
| | photo of a <CLASS>. | 89.17 | 90.59 | 89.87 |
| CoOP [8] | Token 1, 2, 3, 4 | 87.90 | 88.03 | 87.96 |
| | None | 89.25 | 89.44 | 89.34 |
| | Token 1 | 88.62 | 86.29 | 87.44 |
| | Token 2 | 88.99 | 87.01 | 87.99 |
| | Token 3 | 86.47 | 88.90 | 87.67 |
| | Token 4 | **89.27** | 88.95 | 89.11 |
| | Token 1, 2 | 88.00 | 87.45 | 87.72 |
| | Token 3, 4 | 87.55 | 89.17 | 88.35 |
| | Token 1, 4 | 89.12 | 89.61 | **89.36** |
| | Token 1, 2, 3 | 86.38 | 88.91 | 87.63 |
| | Token 1, 2, 4 | 88.84 | 89.58 | 89.21 |
| | Token 1, 3, 4 | 87.73 | **89.67** | 88.69 |
| | Token 2, 3, 4 | 87.41 | 89.41 | 88.40 |
| CoCoOP [11] | Token 1, 2, 3, 4 | 88.73 | 89.60 | 89.16 |
| | None | 89.25 | 89.44 | 89.34 |
| | Token 1 | 88.77 | 89.55 | 89.16 |
| | Token 2 | 89.31 | 90.45 | 89.87 |
| | Token 3 | **89.57** | **90.83** | **90.20** |
| | Token 4 | 89.01 | 90.40 | 89.70 |
| | Token 1, 2 | 88.63 | 89.55 | 89.09 |
| | Token 3, 4 | 89.08 | 90.68 | 89.87 |
| | Token 1, 4 | 88.49 | 89.57 | 89.03 |
| | Token 1, 2, 3 | 88.40 | 89.86 | 89.12 |
| | Token 1, 2, 4 | 88.24 | 89.59 | 88.91 |
| | Token 1, 3, 4 | 88.37 | 90.15 | 89.25 |
| | Token 2, 3, 4 | 88.53 | 90.28 | 89.40 |
| IPO | Identify the primary ingredient in the <CLASS> and describe its texture, color, and presentation. | **89.78** | **91.59** | **90.67** |
| | <CLASS>. | 89.25 | 89.44 | 89.34 |
| | Identify the <CLASS>. | 89.08 | 90.80 | 89.93 |
| | Primary ingredient in the <CLASS>. | 80.55 | 82.15 | 81.34 |
| | Identify the primary ingredient in the <CLASS>. | 88.99 | 90.44 | 89.71 |
| | Describe the <CLASS>'s texture. | 89.32 | 90.39 | 89.85 |
| | Describe the <CLASS>'s color. | 87.75 | 88.82 | 88.28 |
| | Describe the <CLASS>'s presentation. | 88.48 | 89.07 | 88.77 |
| | Texture of the <CLASS>. | 88.86 | 91.15 | 89.99 |
| | Color of the <CLASS>. | 87.90 | 90.31 | 89.09 |
| | Presentation of the <CLASS>. | 88.74 | 90.98 | 89.85 |
| | The <CLASS>'s primary ingredient. | 87.44 | 87.89 | 87.66 |
| | Identify the primary ingredient in the <CLASS> and describe its texture. | 89.35 | 91.06 | 90.20 |
| | Identify the primary ingredient in the <CLASS> and describe its color. | 88.56 | 90.60 | 89.57 |
| | Identify the primary ingredient in the <CLASS> and describe its presentation. | 89.77 | 91.51 | 90.63 |

**Table 24:** Comparison of various prompts with occlusion sensitivity analysis across different models on the Food101 dataset.

| Models | Prompts | Base | Novel | H |
|---|---|---|---|---|
| CLIP [30] | a photo of a <CLASS>. | **27.73** | 33.17 | **30.21** |
| | <CLASS>. | 25.81 | 31.43 | 28.34 |
| | a <CLASS>. | 25.81 | 30.89 | 28.12 |
| | photo <CLASS>. | 22.51 | 30.77 | 26.00 |
| | of <CLASS>. | 24.01 | 30.95 | 27.04 |
| | a photo <CLASS>. | 26.71 | 31.37 | 28.85 |
| | photo of <CLASS>. | 25.03 | 31.97 | 28.08 |
| | of a <CLASS>. | 24.55 | 30.41 | 27.17 |
| | a photo of <CLASS>. | 26.59 | 33.11 | 29.49 |
| | photo of a <CLASS>. | 27.01 | **33.23** | 29.80 |
| CoOP [8] | Token 1, 2, 3, 4 | **27.77** | 27.60 | 27.68 |
| | None | 25.81 | 31.43 | 28.34 |
| | Token 1 | 22.27 | 26.69 | 24.28 |
| | Token 2 | 13.21 | 19.38 | 15.71 |
| | Token 3 | 22.45 | 26.81 | 24.44 |
| | Token 4 | 23.41 | 28.31 | 25.63 |
| | Token 1, 2 | 12.91 | 12.90 | 12.90 |
| | Token 3, 4 | 27.61 | 33.11 | 30.11 |
| | Token 1, 4 | 23.65 | 30.17 | 26.52 |
| | Token 1, 2, 3 | 17.95 | 18.72 | 18.33 |
| | Token 1, 2, 4 | 18.49 | 20.82 | 19.59 |
| | Token 1, 3, 4 | 27.01 | **35.27** | **30.59** |
| | Token 2, 3, 4 | 27.67 | 28.73 | 28.19 |
| CoCoOP [11] | Token 1, 2, 3, 4 | 29.77 | 31.23 | 30.28 |
| | None | 25.81 | 31.43 | 28.34 |
| | Token 1 | 25.93 | 33.17 | 29.11 |
| | Token 2 | 26.59 | 32.33 | 29.18 |
| | Token 3 | 25.75 | 30.65 | 27.99 |
| | Token 4 | 29.23 | 33.17 | 31.08 |
| | Token 1, 2 | 28.45 | 32.87 | 30.50 |
| | Token 3, 4 | 30.43 | 33.65 | 31.96 |
| | Token 1, 4 | **31.33** | 35.21 | 33.16 |
| | Token 1, 2, 3 | 28.75 | 31.49 | 30.06 |
| | Token 1, 2, 4 | **31.33** | **35.81** | **33.42** |
| | Token 1, 3, 4 | 30.19 | 35.51 | 32.63 |
| | Token 2, 3, 4 | 30.13 | 35.45 | 32.57 |
| IPO | Capture a comprehensive range of well-lit, high-resolution images of an <CLASS> from various angles, meticulously showcasing its specific design features with perfect clarity and precision for unparalleled accuracy in aircraft. | **31.43** | **36.32** | **33.70** |
| | <CLASS>. | 25.81 | 31.43 | 28.34 |
| | Capture an <CLASS>. | 26.53 | 31.79 | 28.92 |
| | Range of images of an <CLASS>. | 28.81 | 32.93 | 30.73 |
| | Well-lit images of an <CLASS>. | 26.53 | 32.75 | 29.31 |
| | High-resolution images of an <CLASS>. | 28.99 | 33.11 | 30.91 |
| | Images of an <CLASS> from various angles. | 28.21 | 34.13 | 30.89 |
| | Comprehensive range of images of an <CLASS>. | 28.21 | 32.75 | 30.31 |
| | Meticulously showcase the <CLASS>'s design features. | 25.03 | 30.83 | 27.63 |
| | Specific design features of an <CLASS>. | 28.27 | 34.01 | 30.88 |
| | Perfect clarity and precision of the <CLASS>. | 23.77 | 31.31 | 27.02 |
| | Capture well-lit images of an <CLASS>. | 27.31 | 30.71 | 28.91 |
| | Capture high-resolution images of an <CLASS>. | 30.07 | 32.81 | 31.38 |
| | Capture images of an <CLASS> from various angles. | 27.79 | 32.63 | 30.02 |
| | Showcase the <CLASS> with clarity and precision. | 25.27 | 22.26 | 23.67 |
| | Unparalleled accuracy of the <CLASS> in aircraft. | 28.69 | 33.05 | 30.72 |

**Table 25:** Comparison of various prompts with occlusion sensitivity analysis across different models on the FGVCAircraft dataset.

| Models | Prompts | Base | Novel | H |
|---|---|---|---|---|
| CLIP [30] | a photo of a <CLASS>. | 69.36 | 75.35 | 72.23 |
| | <CLASS>. | 65.91 | 72.74 | 69.16 |
| | a <CLASS>. | 68.75 | 66.65 | 67.68 |
| | photo <CLASS>. | 62.88 | 70.42 | 66.44 |
| | of <CLASS>. | 65.27 | 69.95 | 67.53 |
| | a photo <CLASS>. | 65.01 | **75.73** | 69.96 |
| | photo of <CLASS>. | **70.47** | 72.75 | 71.59 |
| | of a <CLASS>. | 69.19 | 75.47 | 72.19 |
| | a photo of <CLASS>. | 69.76 | 75.55 | **72.54** |
| | photo of a <CLASS>. | 69.52 | 70.62 | 70.07 |
| CoOP [8] | Token 1, 2, 3, 4 | **71.47** | 72.47 | **71.97** |
| | None | 65.91 | **72.74** | 69.16 |
| | Token 1 | 61.39 | 65.98 | 63.60 |
| | Token 2 | 61.14 | 66.43 | 63.68 |
| | Token 3 | 69.87 | 71.54 | 70.70 |
| | Token 4 | 63.72 | 66.22 | 64.95 |
| | Token 1, 2 | 64.76 | 68.34 | 66.50 |
| | Token 3, 4 | 69.51 | 71.57 | 70.52 |
| | Token 1, 4 | 65.23 | 68.66 | 66.90 |
| | Token 1, 2, 3 | 69.98 | 72.66 | 71.29 |
| | Token 1, 2, 4 | 66.27 | 72.07 | 69.05 |
| | Token 1, 3, 4 | 69.64 | 71.45 | 70.53 |
| | Token 2, 3, 4 | 69.62 | 72.22 | 70.90 |
| CoCoOP [11] | Token 1, 2, 3, 4 | **73.67** | 75.50 | **74.57** |
| | None | 65.91 | 72.74 | 69.16 |
| | Token 1 | 70.85 | **75.63** | 73.16 |
| | Token 2 | 70.52 | 75.25 | 72.81 |
| | Token 3 | 69.64 | 75.48 | 72.44 |
| | Token 4 | 71.64 | 75.05 | 73.31 |
| | Token 1, 2 | 72.55 | 72.65 | 72.60 |
| | Token 3, 4 | 73.56 | 72.61 | 73.08 |
| | Token 1, 4 | 73.18 | 72.72 | 72.95 |
| | Token 1, 2, 3 | 72.53 | 69.01 | 70.73 |
| | Token 1, 2, 4 | 73.26 | 68.87 | 71.00 |
| | Token 1, 3, 4 | 73.46 | 68.75 | 71.03 |
| | Token 2, 3, 4 | 73.24 | 68.02 | 70.53 |
| IPO | A photo of a <CLASS>, a type of large-scale scene. | **72.25** | **77.53** | **74.80** |
| | <CLASS>. | 65.91 | 72.74 | 69.16 |
| | A photo of a <CLASS>. | 69.36 | 75.35 | 72.23 |
| | Photo of a <CLASS>. | 69.52 | 70.62 | 70.07 |
| | A <CLASS>, a type of large-scale scene. | 71.51 | 77.02 | 74.16 |
| | Large-scale scene of a <CLASS>. | 71.03 | 76.63 | 73.72 |
| | Type of large-scale scene: <CLASS>. | 69.97 | 76.08 | 72.90 |
| | A photo of the <CLASS>. | 71.74 | 76.71 | 74.14 |
| | The <CLASS>, a large-scale scene. | 69.83 | 74.83 | 72.24 |
| | A type of large-scale scene: <CLASS>. | 70.92 | 76.64 | 73.67 |
| | Scene of a <CLASS>. | 70.55 | 74.95 | 72.68 |

**Table 26:** Comparison of various prompts with occlusion sensitivity analysis across different models on the SUN397 dataset.

| Models | Prompts | Base | Novel | H |
|---|---|---|---|---|
| CLIP [30] | a photo of a <CLASS>. | 54.63 | 59.18 | 56.81 |
| | <CLASS>. | 53.70 | 59.18 | 56.31 |
| | a <CLASS>. | 53.70 | 60.27 | 56.80 |
| | photo <CLASS>. | 43.63 | 44.93 | 44.27 |
| | of <CLASS>. | **55.79** | 58.82 | 57.26 |
| | a photo <CLASS>. | 43.17 | 41.43 | 42.28 |
| | photo of <CLASS>. | 53.47 | 52.05 | 52.75 |
| | of a <CLASS>. | 55.21 | **60.75** | **57.85** |
| | a photo of <CLASS>. | 53.36 | 51.69 | 52.51 |
| | photo of a <CLASS>. | 54.63 | 59.78 | 57.09 |
| CoOP [8] | Token 1, 2, 3, 4 | **60.80** | 47.53 | 53.35 |
| | None | 53.70 | **59.18** | **56.31** |
| | Token 1 | 50.93 | 52.66 | 51.78 |
| | Token 2 | 49.88 | 48.07 | 48.96 |
| | Token 3 | 59.03 | 44.93 | 51.02 |
| | Token 4 | 54.05 | 55.43 | 54.73 |
| | Token 1, 2 | 46.53 | 42.03 | 44.17 |
| | Token 3, 4 | 55.90 | 46.74 | 50.91 |
| | Token 1, 4 | 54.17 | 57.00 | 55.55 |
| | Token 1, 2, 3 | 54.51 | 48.67 | 51.42 |
| | Token 1, 2, 4 | 51.62 | 44.69 | 47.91 |
| | Token 1, 3, 4 | 55.67 | 46.74 | 50.82 |
| | Token 2, 3, 4 | 55.32 | 46.38 | 50.46 |
| CoCoOP [11] | Token 1, 2, 3, 4 | 58.70 | 52.70 | 55.54 |
| | None | 53.70 | **59.18** | **56.31** |
| | Token 1 | 57.41 | 51.45 | 54.27 |
| | Token 2 | 57.64 | 50.60 | 53.89 |
| | Token 3 | **59.03** | 53.38 | 56.06 |
| | Token 4 | 57.87 | 52.66 | 55.14 |
| | Token 1, 2 | 55.44 | 52.05 | 53.69 |
| | Token 3, 4 | 57.06 | 54.83 | 55.92 |
| | Token 1, 4 | 56.37 | 53.62 | 54.96 |
| | Token 1, 2, 3 | 56.02 | 49.28 | 52.43 |
| | Token 1, 2, 4 | 55.44 | 52.29 | 53.82 |
| | Token 1, 3, 4 | 57.29 | 52.90 | 55.01 |
| | Token 2, 3, 4 | 57.18 | 52.78 | 54.89 |
| IPO | Classify the intricate <CLASS> texture. | 55.45 | 62.47 | **58.75** |
| | <CLASS>. | 53.70 | 59.18 | 56.31 |
| | Classify <CLASS>. | 53.01 | 57.97 | 55.38 |
| | The intricate <CLASS>. | 54.17 | 58.70 | 56.34 |
| | Intricate <CLASS> texture. | 52.66 | 55.43 | 54.01 |
| | Classify the <CLASS>. | **55.56** | 59.66 | 57.54 |
| | Classify the texture of the <CLASS>. | 53.70 | 62.44 | 57.74 |
| | <CLASS> texture. | 53.12 | 60.51 | 56.57 |
| | Texture of the <CLASS>. | 52.43 | **63.77** | 57.55 |
| | The <CLASS>'s intricate texture. | 52.31 | 60.63 | 56.16 |
| | Classify intricate <CLASS> texture. | 54.40 | 58.70 | 56.47 |

**Table 27:** Comparison of various prompts with occlusion sensitivity analysis across different models on the DTD dataset.

| Models | Prompts | Base | Novel | H |
|---|---|---|---|---|
| CLIP [30] | a photo of a <CLASS>. | 50.26 | 69.90 | 58.47 |
| | <CLASS>. | 47.19 | 66.49 | 55.20 |
| | a <CLASS>. | 53.57 | 70.64 | 60.93 |
| | photo <CLASS>. | 46.26 | 53.77 | 49.73 |
| | of <CLASS>. | 54.14 | **74.23** | 62.61 |
| | a photo <CLASS>. | 48.14 | 62.95 | 54.56 |
| | photo of <CLASS>. | 48.02 | 67.92 | 56.26 |
| | of a <CLASS>. | **62.43** | 71.31 | **66.58** |
| | a photo of <CLASS>. | 49.62 | 67.97 | 57.36 |
| | photo of a <CLASS>. | 48.43 | 67.38 | 56.35 |
| CoOP [8] | Token 1, 2, 3, 4 | 69.13 | 50.33 | 58.25 |
| | None | 47.19 | 66.49 | 55.20 |
| | Token 1 | 47.95 | 67.85 | 56.19 |
| | Token 2 | 58.83 | 66.03 | 62.22 |
| | Token 3 | 74.88 | 46.00 | **69.55** |
| | Token 4 | 57.30 | **71.59** | 63.65 |
| | Token 1, 2 | 57.36 | 67.23 | 61.90 |
| | Token 3, 4 | 76.57 | 46.23 | 57.65 |
| | Token 1, 4 | 54.76 | 68.77 | 60.97 |
| | Token 1, 2, 3 | 67.14 | 40.36 | 50.41 |
| | Token 1, 2, 4 | 61.67 | 65.51 | 63.53 |
| | Token 1, 3, 4 | 75.40 | 45.31 | 56.60 |
| | Token 2, 3, 4 | **79.14** | 44.77 | 57.19 |
| CoCoOP [11] | Token 1, 2, 3, 4 | 71.13 | 62.87 | **66.75** |
| | None | 47.19 | **66.49** | 55.20 |
| | Token 1 | 74.62 | 51.38 | 60.86 |
| | Token 2 | 72.83 | 50.15 | 59.40 |
| | Token 3 | 76.05 | 48.85 | 59.49 |
| | Token 4 | 71.21 | 50.10 | 58.82 |
| | Token 1, 2 | 72.10 | 47.95 | 57.60 |
| | Token 3, 4 | 72.55 | 58.23 | 64.61 |
| | Token 1, 4 | **76.26** | 58.92 | 66.48 |
| | Token 1, 2, 3 | 74.67 | 40.21 | 52.27 |
| | Token 1, 2, 4 | 75.36 | 51.28 | 61.03 |
| | Token 1, 3, 4 | 70.40 | 51.08 | 59.20 |
| | Token 2, 3, 4 | 66.83 | 47.54 | 55.56 |
| IPO | Analyze the <CLASS> vehicles in the satellite image with state-of-the-art algorithms for precise classification and optimal efficiency. | **64.97** | **82.13** | **72.54** |
| | <CLASS>. | 47.19 | 66.49 | 55.20 |
| | Analyze <CLASS> vehicles. | 48.95 | 74.46 | 59.07 |
| | Analyze the <CLASS>. | 51.60 | 75.64 | 61.35 |
| | <CLASS> vehicles in the satellite image. | 51.55 | 80.15 | 62.74 |
| | Vehicles in the satellite image: <CLASS>. | 58.26 | 69.67 | 63.46 |
| | State-of-the-art algorithms for <CLASS> classification. | 54.60 | 74.85 | 63.14 |
| | Precise classification of <CLASS> vehicles. | 61.71 | 76.59 | 68.35 |
| | Optimal efficiency in <CLASS> analysis. | 56.12 | 79.26 | 65.71 |
| | Analyze the <CLASS> vehicles in the satellite image. | 58.17 | 78.82 | 66.94 |
| | Use state-of-the-art algorithms for <CLASS> classification. | 58.17 | 75.54 | 65.73 |
| | Classification and efficiency of <CLASS> vehicles. | 63.79 | 76.46 | 69.55 |

**Table 28:** Comparison of various prompts with occlusion sensitivity analysis across different models on the EuroSAT dataset.

| Models | Prompts | Base | Novel | H |
|---|---|---|---|---|
| CLIP [30] | a photo of a <CLASS>. | 68.15 | 75.07 | 71.44 |
| | <CLASS>. | 67.68 | 72.47 | 67.00 |
| | a <CLASS>. | 68.25 | 69.39 | 68.82 |
| | photo <CLASS>. | 65.05 | 68.74 | 66.84 |
| | of <CLASS>. | 68.87 | 72.31 | 70.55 |
| | a photo <CLASS>. | 64.74 | 71.98 | 68.17 |
| | photo of <CLASS>. | **69.70** | **77.23** | **73.27** |
| | of a <CLASS>. | 68.20 | 71.01 | 69.58 |
| | a photo of <CLASS>. | 68.61 | 76.10 | 72.16 |
| | photo of a <CLASS>. | 68.10 | 75.01 | 71.39 |
| CoOP [8] | Token 1, 2, 3, 4 | 72.50 | 63.57 | 67.74 |
| | None | 67.68 | 72.47 | 67.00 |
| | Token 1 | 65.21 | 71.55 | 68.23 |
| | Token 2 | 64.32 | 70.90 | 67.45 |
| | Token 3 | 74.20 | 71.39 | 72.77 |
| | Token 4 | 65.15 | 70.63 | 67.78 |
| | Token 1, 2 | 64.63 | **73.12** | 68.61 |
| | Token 3, 4 | 72.75 | 69.17 | 70.91 |
| | Token 1, 4 | 66.91 | 70.96 | 68.88 |
| | Token 1, 2, 3 | **74.61** | **73.12** | **73.86** |
| | Token 1, 2, 4 | 64.63 | 72.69 | 68.42 |
| | Token 1, 3, 4 | 73.11 | 69.33 | 71.17 |
| | Token 2, 3, 4 | 74.20 | 69.01 | 71.51 |
| CoCoOP [11] | Token 1, 2, 3, 4 | 74.73 | **72.80** | 73.75 |
| | None | 67.68 | 72.47 | 67.00 |
| | Token 1 | 72.60 | 72.53 | 72.56 |
| | Token 2 | 71.92 | 71.82 | 71.87 |
| | Token 3 | 71.20 | 70.96 | 71.08 |
| | Token 4 | 71.77 | 70.52 | 71.14 |
| | Token 1, 2 | 73.73 | 72.42 | 73.07 |
| | Token 3, 4 | 74.72 | 71.66 | 73.16 |
| | Token 1, 4 | 75.54 | 72.58 | 74.03 |
| | Token 1, 2, 3 | 74.61 | 72.20 | 73.39 |
| | Token 1, 2, 4 | 75.08 | 71.12 | 73.05 |
| | Token 1, 3, 4 | **76.27** | 72.58 | **74.38** |
| | Token 2, 3, 4 | 76.01 | 72.28 | 74.10 |
| IPO | Capture a high-quality, well-lit image of a person flawlessly demonstrating the <CLASS> action with impeccable visual representation to achieve unmatched. | 72.43 | **79.35** | **75.73** |
| | <CLASS>. | 67.68 | 72.47 | 67.00 |
| | Capture a <CLASS>. | 67.32 | 71.23 | 69.22 |
| | High-quality image of a <CLASS>. | 71.41 | 76.96 | 74.08 |
| | Well-lit image of a <CLASS>. | 69.49 | 77.56 | 73.30 |
| | Image of a <CLASS>. | 70.17 | 75.5 | 72.74 |
| | Person demonstrating the <CLASS>. | **72.60** | 76.31 | 74.41 |
| | Demonstrating the <CLASS> action. | 71.41 | 74.96 | 73.14 |
| | Impeccable visual representation of a <CLASS>. | 70.63 | 76.58 | 73.48 |
| | Capture a person demonstrating the <CLASS>. | 70.53 | 76.47 | 73.38 |
| | Flawlessly demonstrating the <CLASS>. | 70.48 | 78.26 | 74.17 |
| | Visual representation of the <CLASS>. | 70.84 | 75.72 | 73.20 |
| | Achieve unmatched representation of a <CLASS>. | 69.03 | 71.55 | 70.27 |

**Table 29:** Comparison of various prompts with occlusion sensitivity analysis across different models on the UCF101 dataset.

