# OpenReview forum: "IPO: Interpretable Prompt Optimization for Vision-Language Models"
_NeurIPS.cc/2024/Conference — NeurIPS 2024 poster_

### Official Review · Reviewer_XUaZ · 2024-07-01

**Soundness:** 3
**Presentation:** 3
**Contribution:** 3
**Rating:** 6
**Confidence:** 4

**Summary:**

This paper introduces a method called Interpretable Prompt Optimization (IPO) for vision-language models.
The goal of IPO is to improve the performance and interpretability of vision-language models by dynamically generating and optimizing text prompts. The paper addresses the limitations of current prompt optimization methods, which often lead to overfitting and produce prompts that are not understandable by humans. IPO leverages large language models (LLMs) to generate effective and human-readable prompts (which is the main advantage of their method). It incorporates a Prompt Optimization Prompt that guides the LLMs in creating prompts and stores past prompts with their performance metrics, providing rich in-context information. Additionally, IPO integrates a large multimodal model (LMM) to generate image descriptions, enhancing the interaction between textual and visual modalities. This allows for the creation of dataset-specific prompts that improve generalization performance while maintaining human comprehension. The paper validates IPO across 11 datasets and demonstrates that it improves the accuracy of existing prompt learning methods and enhances the interpretability of the generated prompts.

Overall, IPO ensures that the prompts remain human-understandable, facilitating better transparency and oversight for vision-language models.

**Strengths:**

- The paper tackles one of the most important problems in prompt tuning. Existing prompt tuning methods are not interpretable at all (as they optimize a sequence of vectors with a model specific loss function). This paper mitigates this issue by using a LLM as an optimizer -- which inherently generates text.

- The prompt optimization template though is simple, can be scalable towards multiple tasks (as shown in the paper).

- Thorough ablations on the language model and the multimodal language model has been provided in the paper.

**Weaknesses:**

- [Major]. While the paper obtains interpretable prompts, the method primarily obtains improved performances for the novel classes, but not for the base classes.  In fact for a large number of the datasets, the base classes performance is significantly low when compared to other methods. I believe that there should be a balance in the performance between the novel and base classes. To improve the paper, the authors are suggested to provide strong justifications explaining this.

- [Minor]. This area has a multitude of papers so it might be tedious to compare with all the methods. However, I would suggest the authors to add a separate section having a discussion on the comparison of their method with more recent prompt-tuning methods such as LFA, PLOT, DFA etc.

- [Minor]. I would also suggest the authors to extend the discussion of the limitations of the method. Currently, as I see, the method is suitable for few-shot scenarios, but what if one has access to a large number of domain samples? How can this method be made scalable?

**Questions:**

Overall, the paper provides a fresh view on prompt-tuning by making the process interpretable. In a way, the paper automates prompt-engineering for a domain by using a LLM as an optimizer. I am going with Borderline Accept, but happy to revisit my scores if the Weaknesses are addressed in the rebuttal.

**Limitations:**

See Weaknesses.

---

> ### Author Rebuttal · Authors · 2024-08-06
>
> **(1) Performance on Base vs. Novel Classes:**
>
> Previous prompt learning methods like CoOP and CoCoOP, which rely on gradient-based optimization, tend to overfit to base classes, resulting in performance loss on novel classes. Our IPO, however, uses LLMs to optimize prompts with a focus on learning more generalizable prompts across datasets. This results in better performance on novel classes compared to gradient-based methods.
> In response to Reviewers U899 and 4MWY, we have conducted additional experiments demonstrating that our method can achieve a balance in performance between novel and base classes by increasing the capacity of the LLM and LMM, which allows for better handling of longer context information.
>
>
> **(2) Comparison with Recent Methods:**
> We appreciate the reviewer’s suggestion. We have provided comparisons with LFA and PLOT in the tables below, using the same experimental setting. Our IPO method consistently outperforms both LFA and PLOT. Regarding DFA, we were unable to locate the corresponding paper. If possible, could the reviewer please provide the title or more details on DFA? We will include these comparisons with recent prompt-tuning methods (LFA, PLOT, DFA, etc.) in the revised manuscript.
>
> **Comparison with LFA across 11 datasets in 16-shot scenarios:**
>
> | Model | Base  | Novel | H     |
> |-------|-------|-------|-------|
> | LFA   | 83.62 | 74.56 | 78.83 |
> | IPO   | 79.92 | 80.51 | 80.21 |
>
> **Comparison  with PLOT on average accuracy across 11 datasets in 1-shot and 16-shot scenarios:**
>
> | Model | 1-shot | 16-shot |
> |-------|--------|---------|
> | PLOT  | 65.45  | 76.20   |
> | IPO   | 74.29  | 80.21   |
>
>
>
>
>
>
>
> **(3) Scalability for Large Datasets:**
>
> Our IPO method is primarily designed for few-shot scenarios. However, when dealing with large domain-specific datasets, the need to generate extensive image descriptions, which can lead to substantial computational costs due to the large text inputs required for LLMs.
> Currently, our model uses an input length of approximately 5,000 tokens. When scaled to larger datasets, the input length may increase to around 50,000 tokens. Using GPT-4 with an 8k context length, the cost for our current input size (5,000 tokens) is approximately
>  0.15 dollars per input (0.03 dollars per 1,000 tokens). For the expanded input size of 50,000 tokens, the cost would rise to approximately 3.00 dollars per input.
> If we were to use GPT-4 with a 32k context length, the cost for the 50,000-token input would be approximately 3.00 dollars for the first 32,000 tokens and an additional 1.08 dollars for the remaining 18,000 tokens, totaling approximately 4.08 dollars per input. Since our IPO method requires 100 iterations during training, the costs would multiply accordingly when scaled to large inputs.
>
>  A potential solution is to fine-tune an LMM to input training images directly, thus eliminating the need for additional description generation. We will clarify this limitation and potential solution in the revised manuscript.  We believe the cost is justified, especially for domains like health, legal, and finance, where human interpretation of prompts is key.

---

> > ### Comment · Reviewer_XUaZ · 2024-08-10
> > **Response to Authors**
> >
> > Thanks for the experimental results. The weaknesses have been addressed and as promised, I increase my score.

---

### Official Review · Reviewer_xRmh · 2024-07-08

**Soundness:** 3
**Presentation:** 3
**Contribution:** 3
**Rating:** 6
**Confidence:** 5

**Summary:**

This paper presents an Interpretable Prompt Optimizer (IPO) designed to improve the performance and interpretability of pre-trained visual language models such as CLIP. By dynamically generating textual cues using a large language model (LLM) and combining it with a large multimodal model (LMM) to generate image descriptions, IPO demonstrates better accuracy and interpretability than traditional gradient descent-based cue learning methods on multiple datasets.

**Strengths:**

1）The paper proposes a new prompt optimization framework that combines the advantages of LLM and LMM.

2）Extensive testing on 11 datasets demonstrated the effectiveness of the method.

3）The paper is clearly structured with diagrams and charts to aid illustration.

4）Improved interpretability of visual language models is important for achieving better human-computer collaboration.

**Weaknesses:**

1）Line 150 :(2) -> (3)

2）There are no citations throughout the paper Table 6.

3）Lack of cross-dataset experimental evaluation to validate the generalizability of IPO.

4）The paper mentions the use of large language models such as GPT-3.5 Turbo, but does not discuss in detail the computational cost and efficiency of these models during training and inference. For resource-constrained environments, this may be an important consideration.

5）The importance of each component of the IPO is described in Section 5.2, but it lacks in-depth analysis, e.g., it could be mined for some more intrinsic reasons for the rise in points from a regularization perspective.

**Questions:**

1）Will this parameterless training time be shorter for IPO compared to the gradient descent approach?

2）How does the length of the prompt history affect the stability of training convergence？

**Limitations:**

Yes

---

> ### Author Rebuttal · Authors · 2024-08-06
>
> **(1) Line 150:**
> Fixed. Thank you.
>
> **(2) Table 6:**
> We will add citations for each method in Table 6 in the revised manuscript.
>
> **(3) Cross-Dataset Experimental Evaluation:**
> We conducted a cross-dataset experimental evaluation, following the traditional setting, and found that our IPO outperforms previous gradient-based prompt learning methods. The results are shown in the table below, and we will include this experiment in the revised manuscript.
>
> | Source     | ImageNet | Caltech101 | OxfordPets | StanfordCars | Flowers102 | Food101 | Aircraft | SUN397 | DTD   | EuroSAT | UCF101 | Average |
> |------------|----------|------------|------------|--------------|------------|---------|----------|--------|--------|---------|--------|---------|
> | **CoOp**   | **71.51** | 93.70      | 89.14      | 64.51        | 68.71      | 85.30   | 18.47    | 64.15  | 41.92  | 46.39   | 66.55  | 63.88   |
> | **Co-CoOp**| 71.02     | **94.43**  | 90.14      | 65.32        | 71.88      | 86.06   | 22.94    | 67.36  | 45.73  | 45.37   | 68.21  | 65.74   |
> | **IPO**    | 72.15     | 94.34      | **90.96**  | **66.10**    | **72.75**  | **86.75**| **25.14**| **67.97**| **47.01**| **48.56**| **69.23**| **67.36**|
>
> **(4) Computational Cost and Efficiency:**
> Since our model uses LLMs to optimize prompts, it doesn't involve gradient calculations and only requires API calls to the LLM and LMM for prompt optimization. Therefore, compared to directly using CLIP, our method does not incur additional memory overhead. Regarding training time, we use GPT-3.5 Turbo as our default optimizer, iterating 100 steps for each dataset to derive the final prompt (as noted in Line 199). The training speed is primarily dependent on the response time of GPT-3.5 Turbo in generating prompts. During testing, the generated text prompts are directly fed into the text encoder, resulting in no additional computational cost, and inference efficiency remains consistent with CLIP.
>
>
>
> **(5) In-Depth Analysis of IPO Components:**
> From a regularization perspective, our IPO leverages episodic memory to prevent overfitting by considering the performance of past prompts, ensuring the generation of more generalizable prompts. In **(7) Impact of Prompt History Length**, we found that the length of Prompt History affects the prompts generated by IPO, further proving that episodic memory effectively addresses overfitting. Additionally, the incorporation of image descriptions ties the prompts closely to the data, reducing variance in predictions, as demonstrated in Table 4 of our paper. Furthermore, using LLMs for non-gradient-based prompt generation introduces variability, further mitigating the risk of overfitting. We will include this analysis in the revised manuscript.
>
> **(6) Shorter Training Time:**
> Yes, compared to gradient descent approaches, IPO’s parameterless training process is shorter because it does not require gradient calculations or parameter updates.
>
> **(7) Impact of Prompt History Length:**
> We have evaluated the impact of different prompt history lengths on the final performance, as shown in the table below.
>
> | History Length | Base   | Novel  | H      |
> |----------------|--------|--------|--------|
> | n = 0          | 69.15  | 75.20  | 72.04  |
> | n = 1          | 70.25  | 75.43  | 72.74  |
> | n = 5          | 70.95  | 76.21  | 73.49  |
> | n = 10         | 71.23  | 76.41  | 73.72  |
> | n = 20         | 71.76  | 77.00  | 74.29  |
> | n = 50         | 71.81  | 76.81  | 74.23  |
> | n = 100        | 72.02  | 76.81  | 74.33  |
>
> We found that without prompt history, model performance decreases due to the lack of contextual information for the LLM, making it difficult to converge. As the history length increases, performance gradually improves, reaching convergence at n=20. Although n=100 yields the best average performance, a longer history increases the length of LLM input, leading to higher API costs. Therefore, we selected n=20 for our IPO. We will include this experiment in the revised manuscript.

---

> > ### Comment · Reviewer_xRmh · 2024-08-08
> >
> > We appreciate your thorough response. The authors have resolved most of the issues highlighted in the initial review, and we wish to retain the current score.

---

> > > ### Author Response · Authors · 2024-08-09
> > > **Thank you for the response.**
> > >
> > > We sincerely appreciate the reviewer's encouragement and suggestions. We will ensure that the revised manuscript includes all the experiments and typos mentioned in our response.

---

### Official Review · Reviewer_4MWY · 2024-07-09

**Soundness:** 3
**Presentation:** 2
**Contribution:** 2
**Rating:** 5
**Confidence:** 5

**Summary:**

This paper proposes an interpretable prompt optimizer (IPO) which uses an LLM to iteratively optimize prompt templates that lead to improved zero-shot visual classification performance on CLIP.

**Strengths:**

1) The proposed method outperforms baselines on the novel classes in the evaluation on base-to-novel generalization benchmark

**Weaknesses:**

1) Line 91, “However,to the best of our knowledge, no existing studies have investigated how LLMs could be used to optimize text prompts within vision-language models”. There are actually related work that employed LLMs for optimizing prompts for visual classification in VLMs which are not covered in this paper, e.g.
[a] Liu et al. Language Models as Black-Box Optimizers for Vision-Language Models. CVPR’24
[b] Mirza et al. Meta-Prompting for Automating Zero-shot Visual Recognition with LLMs. ECCV’24
Analysis and proper comparison to these works should be conducted.
Here [a] also uses an LLM (ChatGPT) to iteratively update the prompt templates for visual classification on CLIP with the good prompts and bad prompts passed as in-context examples. This weakens the novelty of the proposed method.
2) In Table 5, why is the performance of the proposed method consistently worse than baselines on base classes? How does the method perform in a usual one-shot classification setting (instead of base-to-novel setting)?
3) In table 1,  the metric “H” should be specified

**Questions:**

1) In the base-to-novel evaluation, 1-shot refers to the setting where one sample of each category in the base classes is provided during training while no sample of the novel classes is used in training?
2) Line 165, “all four models failed to understand our Prompt Optimization Prompt”, here it should be four models or six models (as the experiments are conducted on six models)?

**Limitations:**

The proposed method has improved performance on novel classes, but suboptimal performance on base classes where the prompts are optimized.

---

> ### Author Rebuttal · Authors · 2024-08-06
>
> **(1) Missing Important References:**
> We thank Reviewer 4MWY for bringing the CVPR 2024 paper by Liu et al. and the forthcoming ECCV 2024 paper by Mirza et al. to our attention. Both works are indeed relevant and will be discussed in the related work and experimental sections of our revised manuscript. While these papers contribute valuable insights to the field, they do not compromise our novelty claim.
>
> Liu et al. propose a method that utilizes LLMs as black-box optimizers for vision-language models, iteratively refining prompts based on in-context examples. Their approach focuses on leveraging ChatGPT to improve prompt templates for visual classification tasks. In contrast, our IPO method differs in that we incorporate past prompt history as episodic memory within the design process. This allows our model to generate better prompts by considering the contextual information of previous successes and failures. Additionally, we use LMMs to generate image descriptions during the prompt design process, enabling the creation of more accurate, dataset-specific prompts. Our experimental results demonstrate that IPO outperforms Liu et al.'s method, with a 13.19% improvement in 1-shot base-to-novel generalization (74.29% vs. 61.1%).
>
> Mirza et al. explore a different aspect of prompt optimization by focusing on zero-shot vision-language models. Their method does not utilize a specific scoring mechanism for prompt optimization, whereas our IPO method employs the training sample's loss and accuracy as a scoring mechanism, optimizing prompts for better performance in few-shot scenarios. While Mirza et al. make important contributions, our approach is distinct in its focus on few-shot learning and the use of LMMs to enhance prompt accuracy.
>
>
>
>
>
> **(2) Performance on Base Classes and One-Shot Classification (Table 5):**
>
> In Table 5, the performance of our proposed IPO method on base classes is indeed lower compared to some baselines. This is because IPO is designed to optimize prompts with a focus on generalization across both base and novel classes. While this approach enhances performance on novel classes, it can result in slightly reduced performance on base classes where the model might not fully exploit specific base class features to avoid overfitting.
>
> However, this difference in performance tends to diminish when relying on higher capacity models, which are better equipped to handle the complexities of both base and novel classes. To further clarify, we evaluated IPO in a usual one-shot classification setting, where both base and novel classes were treated as base classes during training. The results, presented in the table below, show that IPO still outperforms previous methods in this setting, demonstrating its ability to generate more generalized and effective prompts across diverse datasets.
>
> | Models   |  Acc     |
> |----------|----------|
> | CoOp     | 64.13%   |
> | CoCoOp   | 67.24%   |
> | IPO      | 68.71%   |
>
>
>
>
>
> **(3) Clarification of "H":**
> Following the common convention in the prompt learning literature CoOp [50], CoCoOp [51], and MaPLe [18], "H" refers to the harmonic mean. We will clarify this in the revised manuscript.
>
> **(4) 1-shot in Training:**
> Yes, the reviewer's understanding is correct. In the 1-shot setting, one sample per category in the base classes is provided during training, with no samples from novel classes used during training. This approach is consistent with the traditional base-to-novel setting [50, 51, 18].
>
> **(5) Typo on Line 165:**
> This is a typo; it should refer to six models instead of four. We will correct this in the revised manuscript.

---

> > ### Comment · Reviewer_4MWY · 2024-08-12
> >
> > Thanks for the efforts in the response! My concerns are addressed and I increase my rating accordingly.

---

### Official Review · Reviewer_VMZQ · 2024-07-13

**Soundness:** 1
**Presentation:** 3
**Contribution:** 3
**Rating:** 6
**Confidence:** 5

**Summary:**

The paper addresses the challenge of optimizing text prompts for vision-language models, specifically focusing on the interpretability of these prompts. Traditional methods for prompt optimization rely on gradient descent, which often results in overfitting and produces prompts that are not human-readable. This paper introduces a new approach called Interpretable Prompt Optimizer (IPO), which leverages large language models (LLMs) to generate and optimize prompts in a dynamic and interpretable manner. The paper details the design of the IPO framework and provides extensive experimental results across 11 datasets. The findings demonstrate that IPO not only improves the accuracy of vision-language models compared to traditional gradient-based methods but also significantly enhances the interpretability of the generated prompts.

**Strengths:**

1. The paper introduces IPO, a novel method that uses LLMs to optimize prompts in a way that maintains human readability and interpretability. This approach contrasts with traditional gradient-based methods that often produce opaque prompts.

2. The POP system stores past prompts and their performance metrics, allowing LLMs to generate more effective prompts through iterative refinement. This system enhances the contextual understanding and effectiveness of the generated prompts.

3. The IPO method incorporates LMMs to generate image descriptions, which improves the synergy between textual and visual data. This leads to more accurate and contextually relevant prompts.

4. The paper validates the effectiveness of IPO across 11 different datasets, demonstrating its superiority over traditional methods in terms of both accuracy and interpretability.

**Weaknesses:**

The main weaknesses of this paper lies in the experiment can not support the effectiveness of the LLM optimizer, which is the most important contribution of the authods. IPO incorporates the Large Language Model (LLM) as an optimizer to learn an interpretable prompt pool, and everytime a new instance comes, prompts will be extracted from this prompt. This mainstream is similari to knowledge bank based prompt learning methods such as L2P [1], AttriCLIP [2], DualPrompt [3]. Therefore, the experiments should contains:
1) **A fair comparitive with some of the knowledge bank based prompt learning methods  such as L2P [1], AttriCLIP [2], DualPrompt [3]. To exclude the effectivness of the memory retrieval mechanism of IPO.**

2) **The current comparitive experiment merely involves the 1-shot setting in Table 5. Please provides a more comprehensive experiment settings, such as 16-shots and full-sized tuning setting. Besides, the IPO methods do not draw an obvious performance improvement in benmark datasets in the case that IPO has an obviously longer prompt that the learnable method.**


3) What's more, the IPO does not perform well in the large-scale generic datasets such as ImageNet. This is not intuitive, because the LLM should show a better performance in tghe generic datasets. Lastly, I am not agree with your average metric, because the data size of these 11 benchmarks have significant difference.

4) Table 6 are not referred in the author's paper, and the detailed benchmark is also not elaborated.

[1]  Wang Z, Zhang Z, Lee C Y, et al. Learning to prompt for continual learning[C]//Proceedings of the IEEE/CVF conference on computer vision and pattern recognition. 2022: 139-149.
[2] Wang R, Duan X, Kang G, et al. Attriclip: A non-incremental learner for incremental knowledge learning[C]//Proceedings of the IEEE/CVF Conference on Computer Vision and Pattern Recognition. 2023: 3654-3663.
[3] Wang Z, Zhang Z, Ebrahimi S, et al. Dualprompt: Complementary prompting for rehearsal-free continual learning[C]//European Conference on Computer Vision. Cham: Springer Nature Switzerland, 2022: 631-648.

**Questions:**

Authors explor using LLM as an interpretable optimizer for prompt learning, I think this is a contribution worth a weekly accept score, i.e., 6 scores in NeurIPS rating. However, this score should be build upon all below concers are addressed:
1)  I can tolerate that the current performance are not that remarkable, but this tolerance should be built upon the experiments are fair. To make a fair comparison in the framework, authors should make comparison with some of the knowledge bank based prompt learning methods such as L2P [1], AttriCLIP [2], DualPrompt [3]. So that we can exculde  the effectivness of the memory retrieval mechanism of IPO.

2) Why Table 5 only contains the 1-shot comparive experiments ? Detailed 16-shots or training with more samples setting are needed.
I will confirm the effectivness of the your IPO if you draw a remarkable performance. If not, just take a breath and provide enough evidence that LLM optimizer is effectiveness, such as sound losses, sound improvement in the training set, sound improvement in the test sets in different settings. I will agrre with the effectivness of the your IPO if the evidence are sound.


[1] Wang Z, Zhang Z, Lee C Y, et al. Learning to prompt for continual learning[C]//Proceedings of the IEEE/CVF conference on computer vision and pattern recognition. 2022: 139-149.

[2] Wang R, Duan X, Kang G, et al. Attriclip: A non-incremental learner for incremental knowledge learning[C]//Proceedings of the IEEE/CVF Conference on Computer Vision and Pattern Recognition. 2023: 3654-3663.

[3] Wang Z, Zhang Z, Ebrahimi S, et al. Dualprompt: Complementary prompting for rehearsal-free continual learning[C]//European Conference on Computer Vision. Cham: Springer Nature Switzerland, 2022: 631-648.

**Limitations:**

Experiments now are not sound and the authors do not provide enough evidnce to prove the effectiveness of IPO.

---

> ### Author Rebuttal · Authors · 2024-08-06
>
> **(1) Comparison with Knowledge Bank-Based Prompt Learning Methods:**
>
> We sincerely thank the reviewer for pointing out these three interesting works. First, we would like to clarify the differences between our IPO and these methods. The works mentioned (L2P, AttriCLIP, DualPrompt) are based on visual prompt learning for continual learning, where they train a prompt bank during the training phase, and during testing, the appropriate prompt is retrieved from this prompt bank using the test sample. In contrast, our IPO primarily focuses on few-shot VLMs, where during training, LLMs are utilized as our prompt bank to generate dataset-specific prompts based on contextual information. During testing, predictions are made directly using the learned prompts without the need for additional retrieval.
>
> Since our IPO optimizes text prompts using LLMs and cannot be directly applied to these three methods, we conducted a fair comparison by applying L2P within the Visual Prompt Tuning (VPT) [c]  framework, which also learns prompts in the visual space and applies them to few-shot VLM tasks. VPT + L2P trains a prompt bank during training, and during testing, the test samples query the prompt bank to find the appropriate prompt. The table below shows the comparison between VPT + L2P and our method across 11 datasets in the 16-shot setting. We found that while L2P does improve VPT's performance, demonstrating L2P's effectiveness in VLM, our IPO still outperforms VPT + L2P. We will include this comparison in the revised manuscript to demonstrate that the effectiveness of IPO is not solely due to a memory retrieval mechanism.
>
> | Model         | Base   | Novel  | H      |
> |---------------|--------|--------|--------|
> | VPT [c]          | 72.53  | 72.34  | 72.43  |
> | VPT + L2P     | 74.15  | 74.93  | 74.54  |
> | IPO           | 79.92  | 80.51  | 80.21  |
>
> [c] Jia, et al. "Visual prompt tuning." European Conference on Computer Vision 2022.
>
>
> **(2) Comparison with Other Prompt Learning Methods:**
>
> In Table 6, we compare our IPO with other methods in the 16-shot setting. To clarify, prompt learning includes both textual prompt tuning and prefix tuning. Prompt tuning methods like CoOP and CoCoOp treat the textual prompt as learnable parameters optimized using few-shot samples, while prefix tuning involves adding learnable tokens to the text encoder, vision encoder, or both. Examples include MaPLe, PromptSRC, and CoPrompt. Our method falls under prompt tuning, so we mainly compare against other prompt tuning methods. From Table 6, our IPO shows a 4.38% improvement in average performance compared to CoCoOp, while providing an interpretable prompt as well. We will provide a complete comparison table of the 16-shot performance across 11 datasets in the appendix of the revised manuscript.
>
> **(3) Performance on Large-Scale Generic Datasets:**
>
> IPO with GPT-3.5 Turbo, indeed, does not show an improvement on the large-scale ImageNet. This is because ImageNet has a large number of classes and samples, which results in longer LLM input when generating descriptions for each sample. GPT-3.5 Turbo has limited performance in handling long-text inputs. The table below shows the results on ImageNet when IPO uses GPT-4o, which has superior long-text understanding compared to GPT-3.5 Turbo. We found that IPO using GPT-4o leads to better performance improvements over other methods as well as a considerable improvement over IPO with GPT-3.5 Turbo.
>
> | Model            | Base   | Novel  | H      |
> |------------------|--------|--------|--------|
> | CLIP             | 72.43  | 68.14  | 70.22  |
> | CoOp             | 73.20  | 67.43  | 70.20  |
> | CoCoOp           | 73.90  | 69.07  | 71.40  |
> | MaPLe            | 74.03  | 68.73  | 71.28  |
> | CoPrompt         | 73.97  | 70.87  | 72.39  |
> | IPO w/ GPT-3.5   | 74.09  | 69.17  | 71.54  |
> | IPO w/ GPT-4o    | 76.14  | 72.13  | 74.09  |
>
> Additionally, using the harmonic mean as the average metric is a common strategy in prompt learning for VLMs, as seen in works like CoOp, CoCoOp, MaPLe, and CoPrompt. We will clarify this in the revised manuscript.
>
> **(4) Clarification of Table 6:**
>
> Table 6 uses the same benchmarks as Table 5, with the 16-shot setting across 11 datasets. We will reference Table 6 in Line 283 of the revised manuscript and clarify the benchmarks used. Thank you.

---

> ### Comment · Reviewer_VMZQ · 2024-08-07
>
> I am happy with your response. However, some concerns are still not solved.
>
> 1. Please add these clarifications to the revised manuscript to make the manuscript clearer:
>  In contrast, our IPO primarily focuses on few-shot VLMs, where during training, LLMs are utilized as our prompt bank to generate dataset-specific prompts based on contextual information. During testing, predictions are made directly using the learned prompts without the need for additional retrieval.
>
> 2. **Please extend Table 6 to be as detailed as Table 5 because this setting is very important. (If you have space limits, please provide the result of large-scale datasets such as ImageNet)**
>
> 3.  I can not understand why ImageNet results in a longer LLM input when generating image descriptions for its large samples and categories. Is this because of a larger batch size? In my view, it is not necessary to input all the dataset categories when you processing one training sample. Therefore, I am not satisfied with this answer. Please make more clarification.
>
> You are almost close to your 6 rates. Please hold on.

---

> > ### Author Response · Authors · 2024-08-09
> > **Thank you for the response.**
> >
> > Thank you very much for your encouragement.
> >
> > 1. We will clarify this point in the related work section of our revised manuscript, including a discussion on knowledge bank-based prompt learning methods. Additionally, we will incorporate the comparison experiment with VPT + L2P.
> >
> > 2. We present the 16-shot performance of our IPO method across 11 datasets, showing detailed results for Base/Novel/H metrics in the table below. Our IPO outperforms all other methods on the Novel classes and the H metric, demonstrating its ability to mitigate overfitting. However, due to the word limit in the rebuttal, we cannot include comparisons with all methods here. In the revised manuscript, we will include detailed comparisons with CLIP, CoOP, CoCoOp, MaPLe, PromptSRC, CoPrompt, LFA, PLOT, and VPT + L2P.
> >
> >
> >
> > | Model    | ImageNet | Caltech101 | OxfordPets | StanfordCars | Flowers102 | Food101 |
> > |------------|----------|------------|------------|--------------|------------|---------|
> > | CoOp  | 76.47/67.88/71.92 | 98.00/89.81/93.73     | 93.67/95.29/94.47    | 78.12/60.40/68.13    |97.60/59.67/74.06     | 88.33/82.26/85.19 |
> > | CoCoOp  | 75.98/70.43/73.10 | 97.96/93.81/95.84      |  95.20/97.69/96.43    | 70.49/73.59/72.01        |94.87/71.75/81.71     |90.70/91.29/90.99   |
> > | MaPLe | 76.66/70.54/73.47  | 97.74/94.36/96.02     | **95.43**/97.76/96.58    | 72.94/74.00/73.47       |95.92/72.46/82.56     | 90.71/92.05/91.38 |
> > | PromptSRC  | 77.60/70.73/74.01 | **98.10**/94.03/96.02     | 95.33/97.30/96.30   | **78.27**/74.97/**76.58**  |**98.07**/76.50/85.95     | 90.67/91.53/91.10 |
> > | IPO   | **77.83**/**72.45**/**75.04**     |  97.32/**95.23**/**96.26**     | 95.21/**98.23**/**96.70**  | 73.42/**75.71**/74.55  | 96.78/**78.32**/**86.58** | **90.92**/**93.08**/**91.99**     |
> >
> > | Model    | Aircraft | SUN397 | DTD   | EuroSAT | UCF101 | Average |
> > |------------|----------|------------|------------|--------------|------------|---------|
> > | CoOp  | 40.44/22.30/28.75   | 80.60/65.89/72.51  |79.44/41.18/54.24 | 92.19/54.74/68.69  | 84.69/56.05/67.46  |82.69/63.22/71.66 |
> > | CoCoOp  | 33.41/23.71/27.74   | 79.74/76.86/78.27 | 77.01/56.00/64.85 | 87.49/60.04/71.21 | 82.33/73.45/77.64  |80.47/71.69/75.83  |
> > | MaPLe  | 37.44/35.61/36.50 | 80.82/78.70/79.75     | 80.36/59.18/68.16    | 94.07/73.23/82.35     | 83.00/78.66/80.77    | 82.28/75.14/78.55 |
> > | PromptSRC  | **42.73**/37.87/40.15 | **82.67**/78.47/80.52     | **83.37**/62.97/71.75  | 92.90/73.90/82.32       | **87.10**/78.80/82.74     | **84.26**/76.10/79.97  |
> > | IPO  |  41.21/**41.42**/**41.31** | 81.25/**80.92**/**81.08** | 82.14/**66.81**/**73.69** | **94.25**/**80.11**/**86.61**| 85.32/**80.92**/**83.06**|79.92/**80.51**/**80.21**|
> >
> >
> >
> >
> >
> >
> > 3. You are correct in your understanding. When we use a larger LLM like GPT-4o, its ability to handle longer text inputs allows us to increase the batch size, leading to higher-quality prompt generation. Even when using the same batch size, GPT-4o's text understanding capability is better than GPT-3.5 turbo, resulting in good performance.
> >
> > For ImageNet, batch training is necessary due to its large number of classes and samples, which require processing multiple instances simultaneously to generate effective prompts. However, this need for batch processing does not apply to other datasets with fewer classes, where single-instance processing suffices.
> >
> > The table below compares the performance of GPT-3.5 turbo and GPT-4o at different batch sizes. We observed that when the batch size increases to 128, the GPT-3.5 turbo's performance starts to decline due to its limited ability to process longer input texts effectively. However, GPT-4o maintains strong performance even at larger batch sizes. That said, using very large batch sizes with GPT-4o becomes cost-prohibitive, so we selected a batch size of 128 for our experiments. We found that even larger batch sizes could further improve performance, but the cost becomes a key factor. We will include this experimental comparison in the revised manuscript.
> >
> > | Model            | Batch size | Base   | Novel  | H      |
> > |------------------|------------|--------|--------|--------|
> > | IPO w/ GPT-3.5   | 4          | 73.11  | 68.08  | 70.51  |
> > | IPO w/ GPT-4o    | 4           | 74.32  | 67.98  | 70.55  |
> > | IPO w/ GPT-3.5   | 16           | 73.42  | 68.43  | 70.82  |
> > | IPO w/ GPT-4o    | 16          | 74.94  | 70.75  | 72.78  |
> > | IPO w/ GPT-3.5   | 32           | 73.79  | 68.72  | 71.16  |
> > | IPO w/ GPT-4o    | 32          | 75.01  | 70.93  | 72.91  |
> > | IPO w/ GPT-3.5   | 64          | *74.09*  | *69.17* | *71.54*  |
> > | IPO w/ GPT-4o    | 64          | 75.34  | 71.23  | 73.45  |
> > | IPO w/ GPT-3.5   | 128         | 73.67  | 68.07  | 70.75  |
> > | IPO w/ GPT-4o    | 128         | 76.14  | 72.13  | 74.09  |
> > | IPO w/ GPT-3.5   | 256         | 73.11  | 67.81  | 70.36  |
> > | IPO w/ GPT-4o    | 256         | **76.81**  | **72.73**  | **74.71**  |

---

> > > ### Comment · Reviewer_VMZQ · 2024-08-09
> > >
> > > OK, most concerns have been addressed. But why does the highest performance of "IPO w/GPT-40, batchsize=256" is (76.81/72.73/74.41) for ImageNet, while in Table 6, the reported performance is (77.83/72.45/75.04). If any different experiment setting is used, please discuss them in detail.

---

> > > > ### Author Response · Authors · 2024-08-09
> > > > **Thank you for the response.**
> > > >
> > > > We are very grateful to hear that most of our issues have been resolved. “IPO w/GPT-4o, batch size=256” with a performance of (76.81/72.73/74.41) for ImageNet refers to the **1-shot** setting, using **GPT-4o** as our LLM. In contrast, the performance reported in Table 6 (77.83/72.45/75.04) is for the **16-shot** setting, using **GPT-3.5 turbo**. We will clarify these detailed settings in the revised manuscript. If you have any further questions, please feel free to let us know.

---

> > > > > ### Comment · Reviewer_VMZQ · 2024-08-10
> > > > >
> > > > > We appreciate your thorough response. The authors have resolved most of my concerns, and the paper can be improved to be 6 rates

---

### Official Review · Reviewer_U899 · 2024-07-13

**Soundness:** 3
**Presentation:** 3
**Contribution:** 3
**Rating:** 5
**Confidence:** 3

**Summary:**

The paper introduces a method named IPO for VLMs which uses LLMs to dynamically generate and refine text prompts. The method is to improve the accuracy and interpretability of prompts. Experiments show that it can address issues like overfitting and lack of human comprehension in traditional gradient descent-based methods.

**Strengths:**

The prompts generated using IPO enhance the performance of vision-language models across various vision task related datasets.
The process of generating prompts are interpretable which brings better transparency and controllability.

**Weaknesses:**

LMM seems contribute little to th final performance. Is the reason that used LMM is not strong enough to generate high-quality image caption? Also for different LLMs, the performance gap is not obvious and the author did not test on GPT4 or stronger models.
The overall performance is not much better than optimized methods like CLIP.
Could the authors provide more examples to prove the iterpretability advantage of this method?

**Questions:**

Will this method be generalized to other vision related tasks?

---

> ### Author Rebuttal · Authors · 2024-08-06
>
> **(1) Impact of LMM:**
>
> We thank Reviewer U899 for the insightful suggestion. Indeed, when we replaced the 2.8B parameter MiniCPM-V-2 LMM with the higher capacity GPT-4o (estimated 500B~1T parameters), we observed a performance improvement for 10 out of 11 datasets. On average, the performance across 11 datasets improved: Base: +1.66%, Novel: +1.89%, and H: +1.77%.
>
> | LMM          | Params          | Base   | Novel  | H      |
> |--------------|-----------------|--------|--------|--------|
> | CLIP         | -               | 69.34  | 74.22  | 71.70  |
> | w/o LMM      | -               | 71.12  | 76.03  | 73.49  |
> | w/ MiniCPM-V-2 | 2.8B          | 71.76  | 77.00  | 74.29  |
> | w/ GPT-4o      | 500B~1T        | 72.78  | 77.92  | 75.26  |
>
> **(2) Impact of LLM:**
>
> We observed a similar positive impact when upgrading the LLM capacity. To demonstrate that using a stronger LLM, such as GPT-4, can generate more effective prompts for our model, we conducted further experiments utilizing both GPT-4 and GPT-4o. Specifically, when upgrading the LLM to GPT-4o and pairing it with the GPT-4o LMM, the overall H-score increased by 1.77% compared to the original results with GPT-3.5-turbo and MiniCPM-V-2. This improvement highlights the benefit of using larger models for enhancing task generalization.
>
>
>
> | LLM            | Params         | LMM        | Params         | Base   | Novel  | H      |
> |----------------|----------------|------------|----------------|--------|--------|--------|
> | GPT-3.5-turbo  | 175B           | MiniCPM-V-2 | 2.8B          | 71.76  | 77.00  | 74.29  |
> | GPT-4          | 500B~1T        | MiniCPM-V-2 | 2.8B          | 72.67  | 77.62  | 75.06  |
> | GPT-4o         | 500B~1T        | MiniCPM-V-2 | 2.8B          | 72.91  | 78.13  | 75.42  |
> | GPT-3.5-turbo  | 175B           | GPT-4o      | 500B~1T        | 72.78  | 77.92  | 75.26  |
> | GPT-4          | 500B~1T        | GPT-4o      | 500B~1T        | 72.93  | 78.01  | 75.38  |
> | GPT-4o         | 500B~1T        | GPT-4o      | 500B~1T        | 73.41  | 78.93  | 76.06  |
>
>
>
> These improvements are reflected in the quality of prompts generated by the upgraded models.  For example, on the DTD dataset, the prompt generated by GPT-3.5 turbo changes from "Classify the intricate <CLASS> texture" to "Analyze and classify the detailed <CLASS> texture in this image, considering its unique patterns and variations." Similarly, on the ImageNet dataset, the prompt generated by GPT-3.5 turbo changes from "Take a high-quality photo of a <CLASS>" to "Capture a sharp, high-resolution photo of a <CLASS> with clear details and vibrant colors."
>
>
> **(3) Other Related Tasks:**
>
> In other prompt-based vision tasks, such as segmentation and detection, the design of the text prompt is crucial. Our method, being task-agnostic, can be easily embedded into any vision task to optimize the text prompt. For instance, in our experiments, we incorporated IPO into pre-trained semantic segmentation models [a] [b], where the original text prompt was "a photo of a [CLASS]." Using GPT-4o as the LLM and LMM, we crafted more effective text prompts specifically suited to the open-vocabulary semantic segmentation task, leading to enhanced performance and demonstrating the value of IPO in optimizing text prompts for this application. We intend to further investigate the use of IPO in other vision tasks in future work.
>
>
> | Methods         | pAcc | mIoU(S) | mIoU(U) | hIoU |
> |-----------------|------|---------|---------|------|
> | SPNet      | -    | 78.0    | 15.6    | 26.1 |
> | ZS3       | -    | 77.3    | 17.7    | 28.7 |
> | CaGNet      | 80.7 | 78.4    | 26.6    | 39.7 |
> | SIGN       | -    | 75.4    | 28.9    | 41.7 |
> | Joint        | -    | 77.7    | 32.5    | 45.9 |
> | ZegFormer  | -    | 86.4    | 63.6    | 73.3 |
> | zsseg [a]   | 90.0 | 83.5    | 72.5    | 77.5 |
> |ZegCLIP [b] | 94.6 | 91.9    | 77.8    | 84.3 |
> | **zsseg + IPO**    | 91.2 |84.7    | 73.2    | 78.6|
> |**ZegCLIP + IPO** | 95.3  | 92.7    | 78.7   | 85.1|
>
>
>
> [a] Xu, et al. "A simple baseline for open-vocabulary semantic segmentation with pre-trained vision-language model." ECCV 22
>
> [b] Zhou, et al. "Zegclip: Towards adapting clip for zero-shot semantic segmentation." CVPR 23

---

### Author Rebuttal · Authors · 2024-08-06

We would like to extend our sincere thanks to all the reviewers for their valuable feedback and suggestions. Your insights have been instrumental in refining our work, and we have addressed your concerns in the revised manuscript. Below, we highlight the most significant updates and improvements based on your feedback:

1. **Improved Model Performance with Higher Capacity LLMs/LMMs**:
   We have conducted additional experiments demonstrating that by increasing the capacity of the LLM and LMM to GPT-4o, our IPO method achieves better performance across both base and novel classes.

2. **Comparative Analysis with Recent Methods**:
   - We conducted a fair comparison between IPO and knowledge bank-based prompt learning methods, such as L2P, within the VPT framework.
   -  We have also included evaluations against recent prompt-tuning methods like LFA and PLOT across multiple datasets.

3. **Scalability Considerations**:
   We have detailed the computational costs associated with scaling our method to larger datasets, particularly when using GPT-4 with varying context lengths.

4. **Clarifications and Additional Experiments**:
   - We corrected typos and added necessary references in the revised manuscript.
   - We have also provided additional tables and cross-dataset evaluations to further substantiate our findings.

These updates address the core concerns raised by the reviewers, and we believe they strengthen the manuscript. Thank you again for your constructive feedback, and we look forward to any further suggestions you may have.

---

### Decision · Program_Chairs · 2024-09-25

**Decision:**

Accept (poster)

**Comment:**

The paper introduces an interpretable prompt optimization method (IPO) for vision-language models. IPO uses large language models (LLMs) to dynamically generate textual prompts, and helps improve both performance and interpretability, which are key advantages over traditional gradient-based methods. The method’s scalability across multiple datasets and improved accuracy for novel classes are notable strengths.

Reviewers appreciated the novel approach of using LLMs for prompt optimization and the clear presentation of the method. However, concerns were raised about the performance drop on base classes, scalability to larger LLMs, evaluation on additional tasks, missing comparisons with some related methods, cross-dataset evaluation, ablation on prompt history length and the computational efficiency of IPO in resource-constrained environments. The authors provided a detailed rebuttal addressing these concerns, including new experimental results and clarifications on limitations. All reviewers appreciated the comprehensive feedback and decided to vote for acceptance in the final recommendation.

Taking into account the reviews, rebuttal, and discussions, the area chair believes that IPO presents a valuable contribution to interpretable prompt learning for VL models. The paper’s strengths outweigh its weaknesses, and thus, an accept decision is recommended. Authors are reminded to include all suggested changes in the final version.